EMBO
reports

# scientific report

# PARP1 orchestrates variant histone exchange in signal-mediated transcriptional activation

*Amanda O'Donnell, Shen-Hsi Yang & Andrew D. Sharrocks*[+]

Faculty of Life Sciences, University of Manchester, Manchester, UK

**Transcriptional activation is accompanied by multiple molecular events that remodel the local chromatin environment in promoter regions. These molecular events are often orchestrated in response to the activation of signalling pathways, as exemplified by the response of immediate early genes such as *FOS* to ERK MAP kinase signalling. Here, we demonstrate that inducible NFI recruitment permits PARP1 binding to the *FOS* promoter by a mutually reinforcing loop. PARP1 and its poly(ADP-ribosyl)ation activity are required for maintaining *FOS* activation kinetics. We also show that the histone variant H2A.Z associates with the *FOS* promoter and acts in a transcription-suppressive manner. However, in response to ERK pathway signalling, H2A.Z is replaced by H2A; PARP1 activity is required to promote this exchange. Thus, our work has revealed an additional facet of PARP1 function in promoting dynamic remodelling of promoter-associated nucleosomes to allow transcriptional activation in response to cellular signalling.**

Keywords: chromatin; H2A.Z; immediate early genes; MAP kinase signalling; PARP

## INTRODUCTION

Transcriptional activation in response to cellular signalling involves the orchestrated integration of molecular activities that affect both chromatin structure and the engagement and activation of RNA polymerase. Studies on immediate early genes such as *FOS* have provided many insights into the underlying molecular mechanisms controlling their activation (as reviewed in Galbraith and Espinosa and O'Donnell *et al*) [1,2]. Signalling through the ERK MAP kinase cascade triggers several sequential molecular events that contribute to IEG activation [2] that ultimately trigger RNA polymerase engagement and transcriptional initiation [3]. Following phosphorylation of the transcription factor ELK1, one of the key initial molecular events at the *FOS* promoter is the enhanced acetylation of the upstream promoter-proximal nucleosome. This in turn leads to greater accessibility of the DNA

in this region, which permits the binding of the transcription factor nuclear factor 1 (NF1), but it remained unclear which molecular processes are driven by NFI [4].

One potential role for NFI might be to recruit in other regulatory proteins that are involved in remodelling promoter architecture; one such prospective candidate is PARP1 [5]. PARP1 catalyses the deposition of poly ADP-ribose (PAR) on substrate proteins with the main recipient being PARP1 itself in an automodification reaction [6]. Although traditionally viewed as being important in the DNA damage response, PARP1 also has a key role in transcriptional regulation. In one mode of action, PARP1 has a positive role in transcriptional activation by evicting the histone demethylase KDM5B and opening of chromatin through eviction of histone H1 [7]. More recently, PARP1 was linked to transcriptional activation in response to LPS signalling by its ability to modify histones with PAR, thereby destabilising their interactions with DNA and enhancing promoter accessibility [8]. PARP1 can therefore participate in transcriptional activation by affecting chromatin structure through a variety of different mechanisms. Reciprocally, PARP1 can be activated on recruitment to chromatin. For example, in *Drosophila melanogaster*, the histone variant H2Av controls both the recruitment and subsequent activation of PARP1 to promote transcriptional activation [9]. Tip60-mediated acetylation of histone H2A can also activate PARP and promote its subsequent spreading across the *Hsp70* locus in response to heat shock [10]. PARP1 also interacts functionally with the ERK pathway, and activated ERK binds and activates the poly(ADP-ribosyl)ation (PARylation) activity of PARP1 [8,11]. A reciprocal role for PARP1 in promoting ERK-dependent ELK1 phosphorylation and activation was also demonstrated [11].

Here, we probed the role of PARP1 in *FOS* promoter activation and demonstrate that PARP1 is rapidly recruited following ERK pathway activation. Molecularly, PARP1 is required for remodelling of the promoter-proximal nucleosome, and orchestrates the exchange of the histone variant H2A.Z with H2A. PARP1 therefore represents a key signal-dependent regulator of *FOS* promoter remodelling and activation.

## RESULTS
### Inducible PARP1 recruitment leads to *FOS* activation

To establish whether PARP1 potentially has a role in *FOS* activation, we first performed chromatin immunoprecipitation

Faculty of Life Sciences, University of Manchester, Michael Smith Building, Oxford Road, Manchester, UK
[+]Corresponding author. Tel: +44 (0)161 275 5979; Fax: +44 (0)161 275 5082; E-mail: a.d.sharrocks@man.ac.uk

(ChIP) analysis in HeLa cells to establish whether PARP1 is present at the *FOS* promoter. Treatment of cells with the ERK pathway activator PMA promoted rapid inducible binding of PARP1 specifically to the *FOS* promoter region as little binding was observed to a distal region (Fig 1A; supplementary Fig S1A online). This binding was dependent on ERK pathway activity, as treatment of cells with the MEK inhibitor U0126 blocked inducible PARP1 binding (Fig 1B). Furthermore, this increased binding was also dependent on the PARylation activity of PARP, as the PARP inhibitor 3AB blocked the increase in PARP1 binding seen on PMA treatment (Fig 1C). Importantly, PARP1 binding was also accompanied by enhanced local PARylation activity as increased PAR could also be detected on the *FOS* promoter following ERK pathway activation (Fig 1D), and again, this signal was dependent on the catalytic activity of PARP (Fig 1E). Inducible PARP1 recruitment to the *FOS* promoter was not limited to HeLa cells but could also be observed in EGF-treated MCF10A cells (supplementary Fig S1D online). Thus active PARP1 is recruited to the *FOS* promoter in response to ERK pathway activity, and its PARylation activity is needed for this recruitment.

To determine whether PARP1 has a role in *FOS* activation, PARP1 levels were depleted by small interfering RNA (siRNA) treatment (supplementary Fig S1E,G online) and the kinetics of *FOS* induction in HeLa cells were monitored. Significant reductions in *FOS* activation at both the messenger RNA (mRNA) (Fig 1F) and protein (supplementary Fig S1H online) levels were observed on PARP1 depletion. In contrast, depletion of PARP2 (supplementary Fig S1F,G online) had no discernible effect on *FOS* activation kinetics (Fig 1F) demonstrating the specificity of action of PARP1. We also compared *Fos* activation levels in wild-type MEFs compared with MEFs from *Parp1*$^{-/-}$ knockout mice. Again significant decreases in the magnitude of *Fos* activation were seen in the absence of PARP1 at both the mRNA and protein levels (Fig 1G; supplementary Fig S1I online). PARP1 was previously shown to promote ERK signalling by enhancing phosphorylation of the transcription factor ELK1 [11]. However, no decreases in the levels of active phosphorylated ERK or ELK1 were observed on PARP1 inhibition or in the *Parp1*$^{-/-}$ cells, indicating that PARP1 acts downstream from these factors (supplementary Fig S1I,L online).

Next we asked whether PARP enzymatic activity is required for the correct *FOS* activation kinetics. We used two different PARP inhibitors, 3AB and KU0058948 and established optimal concentrations at which PARylation activity was fully inhibited (supplementary Fig S1J online). As expected from the siRNA studies, treatment of cells with 3AB initially reduced the levels of *FOS* activation in response to PMA treatment (for example 45 min; Fig 1G). However, unexpectedly, at later time points, hyper-activation was observed (for example, 75 min; Fig 1H) and this translated into increased levels of FOS protein production (supplementary Fig S1L online). Importantly, the same effects were also seen with a different PARP inhibitor, KU0058948 (supplementary Fig S1K online). Thus PARP activity has an additional role in both the activation and shutdown of promoter activity. To further substantiate a role for the catalytic activity of PARP1 in *Fos* activation, we attempted to rescue the reduced *Fos* activation seen in *Parp1* knockout MEFs by re-expression of wild-type PARP1 or the catalytically compromised mutant PARP1(E988K) in these cells (supplementary Fig S1M online).

Wild-type PARP1 restored an increased magnitude of *Fos* activation, but PARP1(E988K) had no effect on *Fos* activation (Fig 1I) thereby further demonstrating the importance of the catalytic activity of PARP1 for promoting maximal *Fos* activation levels.

## NFI and PARP1 mutually promote each other's binding

NFI is inducibly recruited to the *FOS* promoter after activation of the ERK pathway (Fig 2A) [4]. We therefore tested whether NFI was required for PARP1 binding by reducing NFI levels by siRNA treatment. The depletion of NFI substantially reduced the levels of PARP1 recruited to the *FOS* promoter (Fig 2B). As increased binding of both NFI and PARP1 are observed early during the activation process (supplementary Fig S1A,B online), we also tested whether PARP1 depletion reciprocally affected NFI recruitment. Surprisingly, we found that the rapid and transient NFI binding to the *FOS* promoter was greatly reduced on depletion of PARP1 in HeLa cells (Fig 2C). Similarly, the rapid increase in NFI binding to the *Fos* promoter was reduced in *Parp1*$^{-/-}$ MEFs (Fig 2D). This decreased binding is not simply due to a reduction in NFI levels in these knockout MEFs, as NFI levels appear normal (supplementary Fig S2A online). We also tested whether PARylation activity was needed for NFI recruitment and found that treatment with the PARP inhibitors 3AB or PJ-34 reduced NFI binding (Fig 2E; supplementary Fig S2B online). Together these results suggest a model whereby NFI and PARP1 mutually aid each other's binding to the *FOS* promoter. To test this idea further and establish whether NFI and PARP1 might form a complex, we performed co-immunoprecipitation analysis, and found that NFI and PARP1 interact in cells where the ERK pathway has been activated (Fig 2F). Importantly, re-ChIP analysis demonstrated that NFI and PARP1 can be found to co-occupy the *FOS* promoter, consistent with them forming a chromatin-associated complex (Fig 2G).

## PARP1 is required for changes in the chromatin structure

We previously established that ERK-mediated NFI recruitment leads to changes in nucleosomal structure in the *FOS* promoter region [4]. As NFI functionally interacts with PARP1, we tested whether PARP1 might also be involved in eliciting chromatin changes in this region. The levels of H3K9 acetylation are high in the promoter region relative to a neighbouring intergenic region in quiescent cells and are further enhanced in response to PMA stimulation (supplementary Fig S3A online). However, this increase in acetylation levels is blocked on depletion of PARP1. Rapid increases in chromatin accessibility around the '−1' nucleosome are observed within 10 min of PMA induction (supplementary Fig S3B online) [4] and these changes are specific to this region in the *FOS* locus (supplementary Fig S3C online). Importantly however, we found that depletion of PARP1 caused a reduction in the accessibility of the chromatin following PMA treatment (supplementary Fig S3D online). Thus, PARP1 appears important both for the acquisition of chromatin marks associated with gene activation and for allowing increased accessibility of the promoter-proximal nucleosome.

## H2A.Z inhibits *FOS* promoter activity

Genome-wide studies have shown that H2A.Z generally associated with nucleosomes located in and around promoter regions

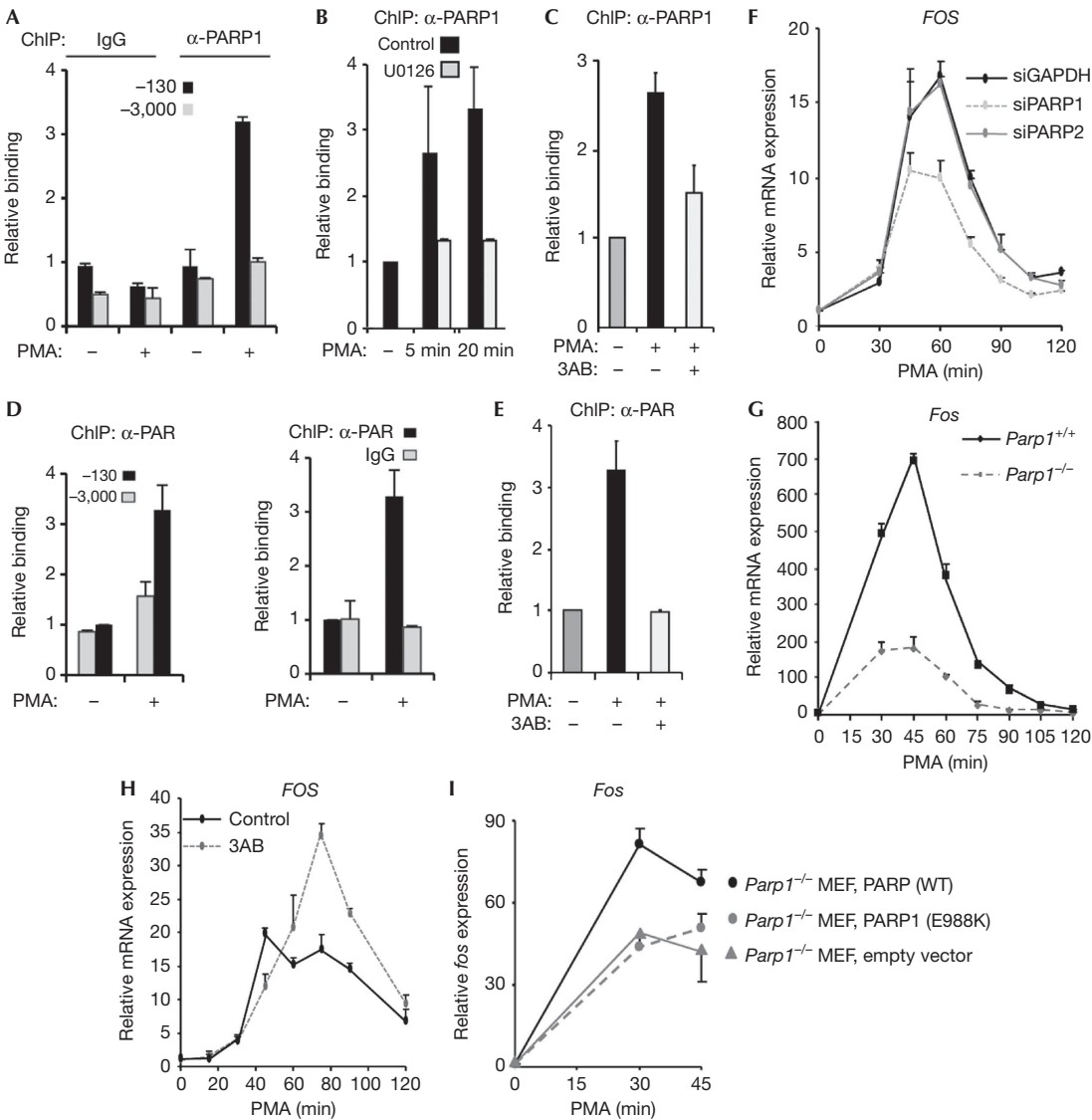

**Fig 1 | PARP1 and PAR are recruited to activate the *FOS* promoter following ERK pathway activation.** ChIP analysis of PARP1 or PAR binding to the *FOS* promoter. (A–E) HeLa cells were starved in serum-free DMEM ( − ) or stimulated with PMA for 10 min ( + ) or for the indicated times. Where indicated, PARP1 inhibitor, 3AB or MEK inhibitor, U0126, was added 30 min before PMA stimulation. Binding was monitored to the promoter (A–E) centred on − 130 bp relative to the TSS or additionally on a control region located at − 3,000 bp (A,D). Data are presented relative to binding in the absence of PMA treatment (taken as 1) as means ± s.e.m. and are the average of at least two independent experiments performed at least in duplicate ($n \geq 4$). (F,G) Real-time RT–PCR measurement of *FOS* mRNA levels in (F) HeLa cells treated with the indicated siRNAs or (G) wild-type MEFs or *Parp1* knockout MEFs after incubation with PMA for the indicated times. (H) Real-time RT–PCR measurement of *FOS* mRNA levels in HeLa cells after incubation with PMA at the time points shown and, where indicated, pre-treated for 30 min with PARP1 inhibitor 3AB. (I) Real-time RT–PCR measurement of *Fos* mRNA levels in *Parp1*$^{-/-}$ MEFs (grey line) or *Parp1*$^{-/-}$ MEFs reconstituted with WT PARP1 (black line) or PARP1(E988K) (grey dashed line) after incubation with PMA for the indicated times. Data in (G) are presented as means ± s.e.m. and are the average of at least two independent experiments performed in duplicate ($n \geq 4$). Data in (I) are presented as means ± s.e.m. ($n = 2$) and data in (F,H) are presented as means ± s.d. and are representative of at least two independent experiments performed in triplicate ($n \geq 3$). ChIP, chromatin immunoprecipitation; DMEM, Dulbecco's modified Eagle medium; MEF, murine embryonic fibroblast; mRNA, messenger RNA; PMA, phorbol myristate acetate; PARP1, Poly [ADP-ribose] polymerase 1; RT–PCR, reverse transcription polymerase chain reaction; siRNA, small interfering RNA; WT, wild type.

and at many genes has a role in transcriptional activation [12]. As PARP1 associates with a highly positioned, regulatory nucleosome in the *FOS* promoter, we asked whether H2A.Z is also found in the *FOS* promoter region and hence might be subject to PARP-mediated regulation.

In the absence of signalling, high H2A.Z levels could be detected in the *FOS* promoter-proximal region relative to the low levels seen at distally located regions (Fig 3A; supplementary Fig S4A online); mononucleosome-specific ChIP provided further evidence that this region coincided with the − 1 nucleosome

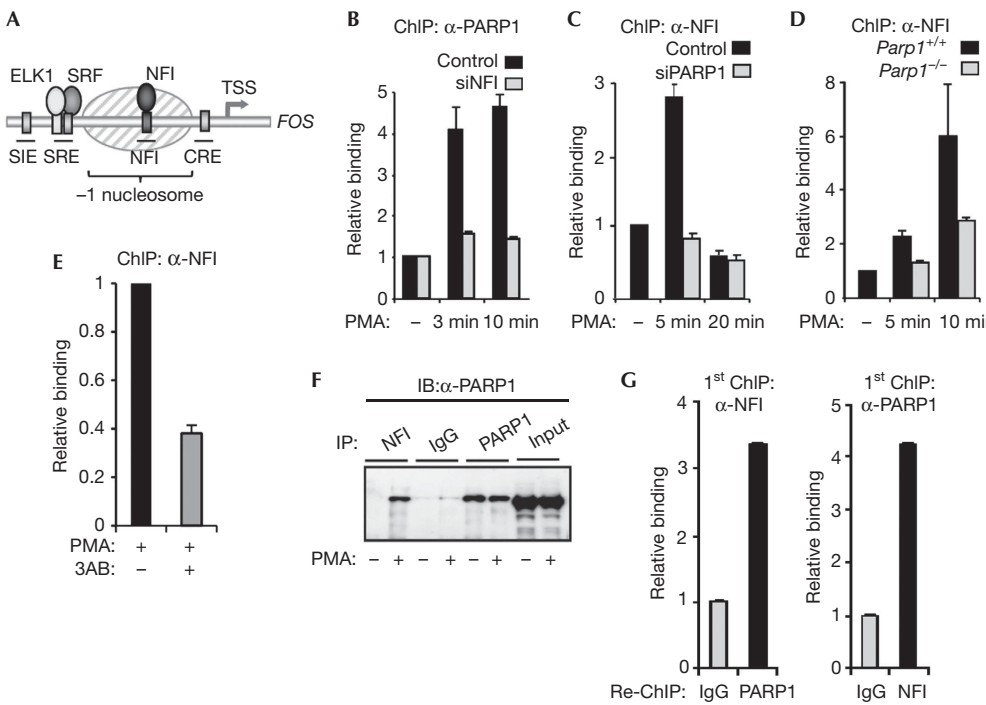

**Fig 2 | PARP1 and NFI mutually aid each other's association with the *FOS* promoter.** (**A**) Schematic diagram of the *FOS* promoter showing the locations of *cis*-regulatory elements. The shaded oval indicates a positioned nucleosome. (**B–E**) ChIP showing the association of PARP1 (**B**) and NFI (**C–E**) with the *FOS* promoter in HeLa cells (**B,C,E**) or wild-type or *parp1*$^{-/-}$ mouse MEFs (**D**). Where indicated, HeLa cells were transfected with siRNAs against NFI, PARP1 or GAPDH; cells were serum-starved ( − ) or starved and treated with PMA for 10 min ( + ) or for times indicated. In (**E**), cells were treated with the PARP inhibitor 3AB for 30 min before PMA stimulation. (**F**) Immunoprecipitations were carried out using NFI, PARP1 or control IgG antibodies followed by immunoblotting (IB for PARP1). HeLa cells were serum-starved ( − ), or starved then stimulated with PMA for 10 min ( + ). (**G**) Re-ChIP analysis of NFI and PARP1 co-association with the *FOS* promoter. HeLa cells were starved and treated with PMA for 10 min. Data in (**B**), (**C**), (**D**) and (**E**) are presented as means ± s.e.m. and are the average of at least two independent experiments performed in at least duplicate (*n* ⩾ 4). Data in (**G**) are presented as means ± s.d. and are representative of at least two independent experiments performed in triplicate (*n* = 3). ChIP, chromatin immunoprecipitation; GAPDH, glyceraldehyde 3-phosphate dehydrogenase; IB, immunoblotting; IgG, immunoglobulin G; MEF, murine embryonic fibroblast; PMA, phorbol myristate acetate; PARP1, Poly [ADP-ribose] polymerase 1; siRNA, small interfering RNA; WT, wild type.

(supplementary Fig S4B,C online). The p400–Tip60 complex is known to have a role in H2A.Z deposition [13], and p400 binding could be detected specifically at the *FOS* promoter region (Fig 3B). Surprisingly, we could also detect INO80 at the *FOS* promoter (Fig 3C), which was unexpected given that INO80 is thought to promote H2A.Z exchange for H2A [14]. This suggested that dynamic changes in the association of p400 and INO80 with the *FOS* promoter might be triggered by PMA treatment. A rapid transient decrease in p400 binding was seen after 5-min treatment of MEFs with PMA (Fig 3D, left panel). Importantly, this decrease mirrored the transient increase in PARylation levels seen at the *Fos* promoter (Fig 3D, right panel) and is consistent with Parp1 promoting p400 loss. To establish whether this was the case, we examined the p400 binding dynamics in *Parp1* knockout MEFs, and no PMA-inducible changes in p400 binding were observed. Ino80 binding increased after 15-min PMA stimulation in wild-type MEFs (Fig 3D, middle panel) but In *Parp1* knockout MEFs, Ino80 binding dynamics changed, with a transient decrease at 10 min, suggesting that Parp1 might be required for maintaining Ino80 at the promoter (Fig 3D, middle panel). These data are therefore consistent with a role for Parp1 in controlling p400/INO80 binding dynamics at the *Fos* promoter. To establish

if this is functionally important, we tested whether INO80 and p400 depletion (supplementary Fig S4E,F,H,I online) affected the levels of H2A and H2A.Z, respectively, at the *FOS* promoter. Depletion of p400 reduced H2A.Z occupancy levels whereas depletion of INO80 caused reductions in H2A occupancy levels as expected from their known regulatory activities (Fig 3E). However, reciprocal effects on the levels of H2A and H2A.Z were not observed.

Next we asked whether H2A.Z and its regulators have a role in determining *FOS* activation kinetics in response to ERK pathway signalling. H2A.Z levels were depleted (supplementary Fig S4D,G online) and this resulted in significant increases in *FOS* activation at the mRNA level (Fig 4A) and in RNA polymerase II associated with the *FOS* gene body (Fig 4B). Moreover, an increase in chromatin accessibility at the − 1 nucleosome was also observed (Fig 4C). Depletion of H2A.Z-deposition enzyme, p400, also resulted in an increase in chromatin accessibility (Fig 4C) suggesting a role for H2A.Z/p400 in creating a transcription-prohibitive chromatin structure at the *FOS* promoter. In comparison, depletion of INO80 has a much smaller effect on chromatin accessibility. Depletion of H2A.Z also enhances NFI binding to the *FOS* promoter (Fig 4D; supplementary Fig S4J online) but,

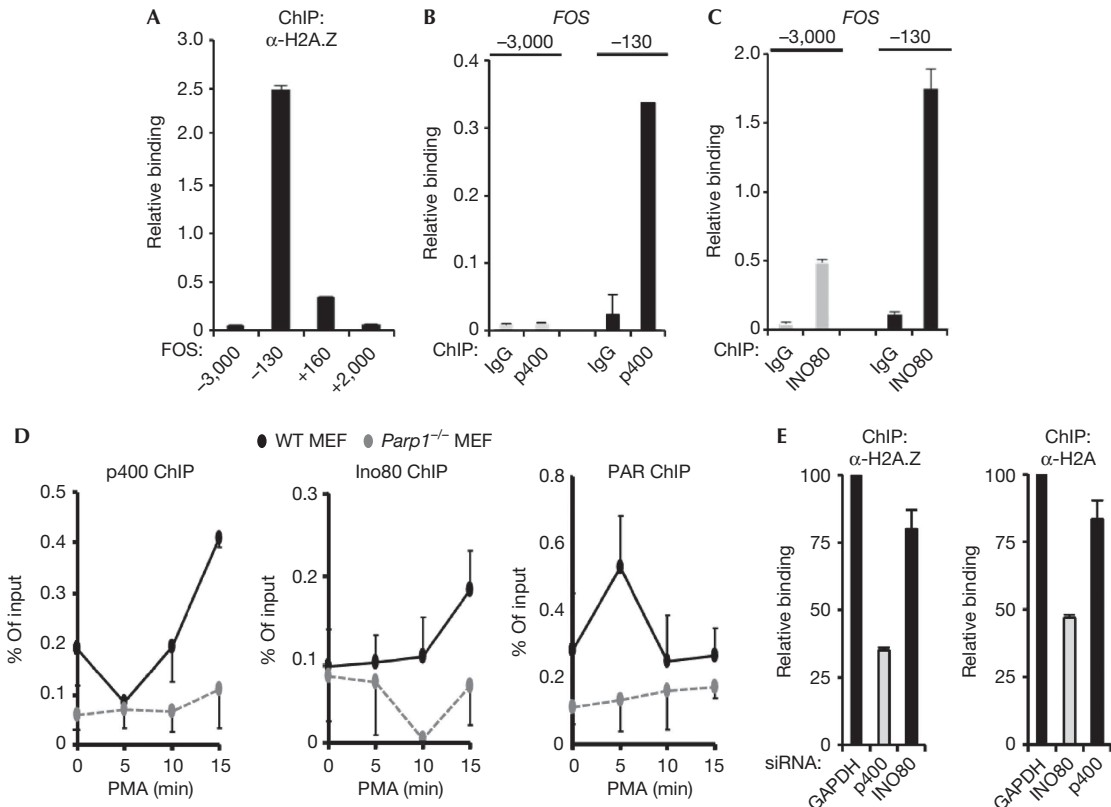

**Fig 3** | H2A.Z localizes to the *FOS* promoter. (**A–C**) ChIP analysis of H2A.Z (**A**), p400 (**B**) and INO80 (**C**) binding to the indicated regions of the *FOS* locus relative to the TSS in quiescent HeLa cells. Binding of H2A.Z is calculated relative to H3 ChIP. Where indicated control ChIP with IgG is shown. (**D**) ChIP in WT or *Parp1* knockout MEFs showing binding of p400, Ino80 and PAR to the *FOS* − 130 region at the indicated times after PMA stimulation. (**E**) ChIP in HeLa cells showing relative binding of H2A and H2A.Z to the *FOS* − 130 region after siRNA-mediated depletion of INO80, p400 or GAPDH as indicated. Data in (**B**), (**C**) and (**E**) are presented as means ± s.e.m. and are the average of at least two independent experiments performed in at least duplicate ($n \geqslant 4$). Data in (**D**) are presented as means ± s.e.m. ($n = 2$) of two independent experiments and data in (**A**) are presented as means ± s.e.m. of duplicate samples and are representative of four independent experiments ($n = 2$). ChIP, chromatin immunoprecipitation; GAPDH, glyceraldehyde 3-phosphate dehydrogenase; IgG, immunoglobulin G; MEF, murine embryonic fibroblast; PMA, phorbol myristate acetate; PARP1, Poly [ADP-ribose] polymerase 1; siRNA, small interfering RNA; TSS, transcription start site; WT, wild type.

comparatively little effect was seen on PARP1 binding (data not shown), indicating that the loss of H2A.Z likely contributes to NFI accessing the *FOS* promoter. Importantly, depletion of p400 increased the levels of *FOS* activation (Fig 4E) whereas depletion of INO80 resulted in reduced levels of *FOS* mRNA expression (Fig 4F) demonstrating that p400 and INO80 have opposite roles in *FOS* gene regulation. Furthermore, in agreement with H2A.Z having a role in inhibiting transcription, the levels of H2A.Z at the *FOS* promoter were substantially reduced following ERK pathway activation, with a concomitant increase in H2A levels (supplementary Fig S4K online). This leads to a rapid decrease in H2A.Z enrichment at the *FOS* promoter, which is initiated within 5 min of PMA stimulation (Fig 4G). Together, these findings demonstrate a role for H2A.Z and its regulators in controlling the levels of *FOS* activation in response to ERK kinase pathway signalling.

## PARP1 is required for histone variant exchange
PARP1 and its associated branched PAR polymer are known to be able to function in transcription as a scaffold, potentially targeting

chromatin modifying enzymes and variant histones to promoter regions [6]. We therefore used co-immunoprecipitation analysis to investigate whether physical associations could be observed between PARP1 and p400 and INO80, and found that PARP1 can form a complex with both of these proteins (Fig 5A). This suggests that PARP1 might have a pivotal role in the homeostatic regulation of dynamic nucleosome reconstruction through inter-action with two opposing ATP-driven enzymes capable of exchanging H2A/H2B and H2A.Z/H2B dimers. Indeed, the rapid PMA-inducible loss of H2A.Z enrichment at the *Fos* promoter seen in wild-type MEFs is compromised in *Parp1* knockout MEFs (Fig 5B; supplementary Fig S5A,B online). Similarly, pharmaco-logical inhibition of PARP1 activity causes a defect in the rapid transient H2A.Z loss at the *FOS* promoter in HeLa cells on PMA stimulation (supplementary Fig S5C online). Furthermore, Parp1 might be important for signal-induced H2A.Z loss in other systems [13], as H2A.Z enrichment levels at the *p21* promoter dropped in wild-type MEFs following treatment with doxyrubicin but were largely maintained in *Parp1* knockout MEFs

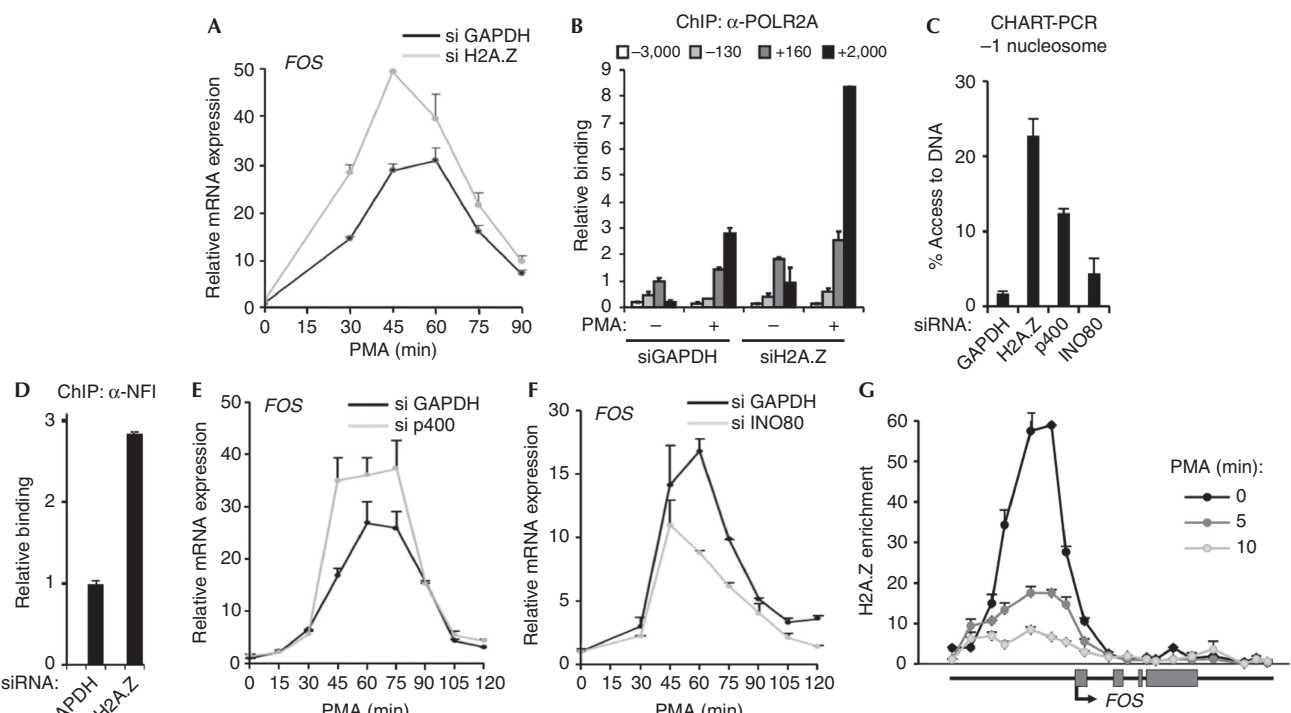

**Fig 4** | H2A.Z negatively regulates *FOS* transcription. (**A**) Real-time RT–PCR measurement of *FOS* mRNA levels in HeLa cells after incubation with PMA for the indicated times in the presence of siRNAs directed towards either GAPDH or H2A.Z. (**B,D**) ChIP showing association of RNA polymerase II (**B**) or NFI (**D**) with the indicated regions of the *FOS* gene. HeLa cells were serum-starved ( − ) or starved and treated with PMA for 5 min ( + ) (**B**) or serum-starved (**D**). Where indicated, cells were pre-treated with siRNA against either GAPDH or H2A.Z. (**C**) CHART–PCR at the *FOS* promoter −1 nucleosome. HeLa cells were treated with PMA for 60 min in cells pre-treated with siRNA against GAPDH, H2A.Z, p400 or INO80. Percentage access was calculated relative to undigested chromatin DNA. (**E,F**) Real-time RT–PCR measurement of *FOS* mRNA levels in HeLa cells after incubation with PMA at the time points shown and, where indicated, siRNAs against GAPDH, p400 or INO80. (**G**) Tiling ChIP analysis of changes in H2A.Z enrichment across the *FOS* locus in serum-starved and after treated with PMA for 5 or 10 min. H2A.Z enrichment is calculated relative to H2A. Data in (**A**), (**B**), (**D**), (**E**) and (**F**) are presented as means ± s.e.m. and are the average of at least two independent experiments performed in at least duplicate ($n \geq 4$). Data in (**C**) and (**G**) are presented as means ± s.d. and are representative of at least two independent experiments performed in triplicate ($n = 3$). ChIP, chromatin immunoprecipitation; GAPDH, glyceraldehyde 3-phosphate dehydrogenase; mRNA, messenger RNA; PMA, phorbol myristate acetate; PARP1, Poly [ADP-ribose] polymerase 1; RT–PCR, reverse transcription polymerase chain reaction; siRNA, small interfering RNA.

(supplementary Fig S5D online). Together, these results point to an important role for PARP1 in controlling the signal-mediated dynamics of H2A.Z incorporation into the promoter regions which in turn leads to promoter activation.

## DISCUSSION

Multiple molecular events have been associated with the transcriptional activation process. These act at both the chromatin level, to change the structure of promoter-associated nucleosomes to allow transcription factors access to genomic information, and at subsequent steps prompting the binding of the transcriptional machinery. Here, we have studied the activation of *FOS*, and demonstrate that PARP1 has a key role in orchestrating localized chromatin-remodelling events in the *FOS* promoter region and allows us to extend our model for how the *FOS* promoter is activated (supplementary Fig S5E online). ERK pathway-mediated inducible NFI recruitment promotes PARP1 binding in a mutually reinforcing manner. Once bound, PARP1, and its associated PARylation activity, facilitate the exchange of repressive H2A.Z

for H2A at the promoter-proximal nucleosome. This establishes a permissive promoter architecture, which enables downstream mediator recruitment and RNA polymerase recruitment and engagement, and hence transcriptional activation.

PARP1 associates with both the p400 and INO80 nucleosome remodelling proteins, and thus has the potential to modify the activities of these proteins, either directly through its interactions or *via* its PARylation activity. Indeed PARP1 can recruit another ATP-dependent chromatin remodeller, CHD1L to chromatin, and subsequent binding to PAR stimulates its helicase activity [15]. As p400 and INO80 are thought to exhibit opposite activities towards H2A.Z deposition [14], an attractive hypothesis is that PARP1 affects H2A.Z binding dynamics through also affecting the activity of one or both of these factors. In the case of p400, PARP1 is needed for transient changes in promoter occupancy, therefore this represents one likely contributory factor to the loss of H2A.Z. However, the interplay between PARP1 and p400 and INO80 is likely complex, and further promoter-associated PARP1 targets might exist. Indeed, ERK signalling can preferentially promote

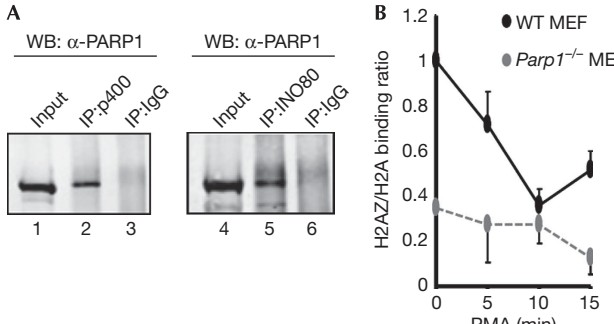

**Fig 5 | PARP1 regulates variant histone exchange. (A)** Co-immunoprecipitation of endogenous PARP1 (detected by WB) using antibodies against non-specific IgG or p400 and INO80 for immunoprecipitation (IP) in HeLa cells. **(B)** ChIP in wild-type (WT) or *Parp1* knockout MEFs showing the relative binding of H2A.Z and H2A to the *FOS* − 130 region at the indicated times after PMA stimulation. Data are presented as a ratio of binding signal (ratio in WT MEFs in the absence of stimulation taken as 1). Data are presented as means ± s.e.m. ($n = 2$) of two independent experiments. ChIP, chromatin immunoprecipitation; IgG, immunoglobulin G; IP, immunoprecipitation; MEF, murine embryonic fibroblast; PMA, phorbol myristate acetate; PARP1, Poly [ADP-ribose] polymerase 1; WB, western blotting; WT, wild type.

ADP-ribosylation of histone H3 [8] raising the possibility that PARP1 might also modify H2A.Z and hence influence its nucleosome-association kinetics.

There are a growing number of links between ERK MAP kinase signalling and PARP1 suggesting that PARP1 will be more widely associated with ERK-mediated gene activation. Similarly, ERK pathway signalling has been shown to contribute to H2A.Z binding dynamics as ERK-mediated transcriptional activation is accompanied by the dissociation of H2A.Z from the *u-PAR* regulatory regions [16]. It is possible that PARP1 might also orchestrate these events on this gene.

In summary, our study adds a new process of histone variant exchange to the growing list of chromatin-associated functions controlled by PARP1 [6]. Importantly, it also extends our understanding of the dynamic events that lead to signal-dependent activation of gene expression. It will be of interest to explore more widely how PARP1 affects inducible gene expression under normal physiological circumstances and in disease situations such as cancer, where ERK pathway signalling is often deregulated.

## METHODS

**Cell culture.** Where indicated, cells were serum-starved for 48 h and either analysed immediately, or stimulated with PMA (10 nM) or EGF (20 ng/ml). When required, cells were pre-treated with MAP kinase inhibitor, UO126 (10 μM), or PARP inhibitors, 3-aminobenzamide (3AB) (2.5 mM), PJ-34 (10 μM) or KU0058948 (250 nM) for 30 min.

**Western Blot and co-immunoprecipitation analysis.** Western blots derived from whole cell lysates or immunoprecipitated proteins were visualized after incubation with primary antibodies using infra-red dye conjugated secondary antibodies and detected by a LiCor Odyssey Infra-red Imager.

**siRNA treatment.** Where indicated, HeLa cells were transfected with 25 nM siRNA targeting *PARP1*, *PARP2*, *H2AZ*, *P400*, *INO80* or *GAPDH* (Dharmacon On Target Plus) using RNAiMAX (Invitrogen). Cells were left for 48 h and then treated with PMA where required.

**ChIP assays and Chromatin Accessibility by Real-Time PCR (CHART–PCR).** ChIP and CHART–PCR assays were carried out essentially as described previously [4].

**Quantitative RT–PCR.** Total RNA was harvested using a RNeasy kit (Qiagen) and transcripts detected in a one-step RT–PCR reaction using Quantitect SYBR green reagent (Qiagen).

**Statistical analysis.** Data are presented as means of a minimum of two biological replicates (where $n = $ an independent biological replicate) and where indicated, one representative experiment (encompassing 2–3 biological repeats) of several experiments is shown. Each datapoint in quantitative RT–PCR and ChIP–PCR experiments is the average of two technical replicates. Where $n = 2$, the source data is also shown in the supplementary information online.

ACKNOWLEDGEMENTS
We thank Karren Palmer for technical assistance; Catherine Millar and members of our laboratory for discussions and comments on the manuscript; Valerie Schreiber and Scott Kaufmann reagents. This work was supported by grants from the Wellcome Trust, Cancer Research UK and a Royal Society-Wolfson award to A.D.S.
    *Author contributions*: A.O'D. and A.D.S. conceived the project and experiments; A.O'D. and S.-H.Y. performed the experiments; A.O'D., S.-H.Y. and A.D.S. wrote the manuscript.

CONFLICT OF INTEREST
The authors declare that they have no conflict of interest.

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
