## [Review Process File · EMBO Reports]

Manuscript EMBOR-2013-37836

PARP1 orchestrates variant histone exchange in signal-mediated transcriptional activation

Amanda O'Donnell, Shen-Hsi Yang, and, Andrew D. Sharrocks

Corresponding author: A.D. Sharrocks, Faculty of Life Sciences, University of Manchester

Review timeline:	Submission to EMBO Journal:	05 October 2012
	Editorial Decision:	12 November 2012
	Revision received:	10 June 2013
	Editorial Decision:	03 July 2013
	Appeal received:	04 July 2013
	Editorial Decision:	04 July 2013
	Submission to EMBO reports:	02 August 2013
	Accepted:	16 September 2013

Transaction Report:

Please note that this manuscript was originally submitted to the EMBO Journal where it was peer-reviewed and revised. It was then transferred to EMBO reports with the original referees' comments and correspondence attached. (Please see below.)

Editors: Anke Sparmann (EMBO Journal) and Esther Schnapp (EMBO reports)

1st Editorial Decision (EMBO Journal)	12 November 2012
------------------

Thank you for submitting your research manuscript (EMBOJ-2012-83512) to our editorial office. It has now been seen by three referees and their comments are provided below.

All reviewers appreciate the concept of your study. However, they raise significant concerns, especially the lack of pertinent controls, and stress that further experiments need to be conducted to clearly demonstrate the validity of your conclusions. In particular, kinetics of the observed phenomena need to be more firmly established, a rescue experiment of PARP1 $-/-$ cells with either wild-type or catalytically inactive PARP should be included, and the model you propose should be tested in in-vitro reconstitution assays.

These issues would have to be addressed by a considerable amount of additional work. As this appears feasible based on the constructive suggestions made by the reviewers, we would be willing to grant the opportunity to significantly extend and revise the current manuscript. This will entail time-consuming experimentation, and therefore we would understand if you might decide to seek rapid publication elsewhere.

However, in case you do embark on revisions for our journal, please take the specified demands into careful consideration to avoid disappointments later in the process. I should add that it is our policy to allow only a single major round of revision and that it is therefore important to address the all raised concerns at this stage.

Thank you for the opportunity to consider your work for publication. I look forward to your revision.

 REFEREE COMMENTS

Referee #1

This manuscript from O'Donnell and Sharrocks describes the role of PARP1 in the activation of the FOS gene. They show that PARP1 is recruited to the FOS promoter (accompanied with an increase in poly-ADP-ribose chains (PAR)) and that depletion of PARP1 decreases by 2 to 3-fold the induction of FOS in response to ERK MAP kinase signalling. They further show that the transcription factor NFI, which they had previously shown to be important for FOS activation, interacts with PARP1 in vivo and participates in its recruitment to FOS. Conversely, depletion of PARP1 also leads to decreased recruitment of NFI. They further characterize the mechanism by which PARP1 regulates FOS expression: depletion of PARP1 reduces acetylation of histone H3K9 and decreases nucleosome accessibility. PARP1 is also shown to interact in vivo with both p400 and Ino80 complexes, which have opposing roles in histone H2A.Z deposition and eviction respectively from the FOS promoter region. Depletion of Ino80 decreases FOS activity, whereas depletion of H2A.Z and p400 increases FOS activity, suggesting a repressive function for p400-mediated H2A.Z deposition. Depletion of PARP1, or inhibition of its activity, affects the kinetics of H2A.Z eviction from the FOS promoter. Altogether, these results lead the authors to suggest a model explaining NFI function in FOS activation kinetics by recruiting PARP, which stimulates the exchange of H2A.Z for H2A, altogether leading to FOS transcription.

General comments:

The authors do a good job at characterizing factors that bind and regulate FOS but several of their conclusions are not really supported by their experiments. The biggest problem is that several effects could simply be attributed to active transcription and not really what the authors claim.

Specific comments:

In figure 3E, the authors use a PARP1 inhibitor, PJ34, to show the function of PARP1 activity for NFI recruitment. Why do the authors use this particular inhibitor here, since two different inhibitors are used in figure 2F? The effect of 3AB and KU0058948 on NFI recruitment should be shown. The effect of PJ34 on FOS transcription should also be shown. If particular inhibitors are not used, it should at least be justified.

In figure 5D, the authors show that a knock-down of p400 decreases H2A.Z and a knock-down of Ino80 decreases H2A at the FOS promoter. p400 has been shown to catalyze the exchange of H2A for H2A.Z, so it would be expected that knock-down should decrease H2A.Z with an increase in H2A. The authors should show a ChIP for H2A in a p400 knock-down and a ChIP for H2A.Z in a Ino80 knock-down.

The authors show that both p400 and Ino80 are present at the FOS promoter (Figure 5B,C). What are the kinetics of p400 and Ino80 after PMA treatment? This is important, since it could explain the kinetics of H2A.Z observed at the promoter.

Then authors then show that PARP1 interacts in vivo with p400 and Ino80 proteins (Figure 7A). An obvious model is that PARP1 or its catalytic activity could function in the recruitment of p400 and Ino80 to the promoter. The authors should perform p400 and Ino80 ChIPs in WT and Parp -/- cells

(or p400 or Ino80 siRNA knock-down cells) or cells treated with 3AB after PMA treatment. This experiment is well within grasp of the authors.

All through the manuscript, the time points taken after PMA treatment for ChIP experiments are inconsistent and no explanation is given. In experiments where a single time point is taken, cells were harvested after either 5min (Figure 6D) or 10 min (other Figures) after PMA treatment. In other experiments, multiple time points were taken at either 5 and 20 min (Figure 1B and 3C), 3 and 10 min (Figure 3B), 5 and 10 min (Figure 3D). This is not trivial, since the dynamics of protein binding to the FOS promoter are very fast. The authors should at least explain these discrepancies.

This point is particularly important in Figure 7B, where H2A.Z is shown to be evicted from the promoter at 5 min, and then returns as soon as 10 min. Then, in Figure 7C, H2A.Z ChIPs are shown for a single time point at 10 min only. Wouldn't it have been more pertinent to perform the ChIP at 5 min, when maximal H2A.Z eviction is expected to be achieved? Given the rapid kinetics, the authors should perform H2A.Z ChIPs at multiple time points after PMA treatment in WT and Parp^{-/-} cells to verify the effect of PARP1 on H2A.Z dynamics.

In Figure S2B, is there a control missing? As I understand the figure, it seems that the effect of PARP2 knock-down on GAPDH is missing.

In Figure S5, the authors perform mono-nucleosome ChIP to verify that the -1 nucleosome is enriched in H2A.Z. Since the cells were fixed with formaldehyde, it is possible that contaminating di-nucleosomes are in the extract and that neighbouring nucleosomes are contributing to the H2A.Z signal. The authors should provide control experiments that were done to ensure minimal di-nucleosome contamination. Also, they could easily pinpoint the location of H2A.Z using PCR probes spanning this region (as in Figure 6G) on the same samples.

Minor comments:

In Figures 4B the CHART-PCR normalization is not clear. How is % access to DNA actually calculated? It is said that 100% is non-digested input DNA. So in that case is access simply (1-% of amplification)? It should be explained in the methods section.

In p.10, line 10. The sentence starting with "Since NFI and PARP1 associate with a highlypositioned..." reads poorly and needs to be revised.

In Figure 5C, I think the axis should say INO80 instead of p400.

In p12, 3rd paragraph. The sentence starting with "Having established that PARP1 is associated..." reads poorly and needs to be revised.

Referee #2

Summary:

In this manuscript, the authors describe a series of experiments examining how the nuclear enzyme PARP-1 affects transcription of the FOS gene during different ERK-pathway controlled stimuli. The authors use molecular biology techniques, such as ChIP, CHART-, and RT-qPCR in conjunction with RNAi-mediated knockdown and chemical inhibition to arrive at a proposed mechanism by which PARP-1 affects FOS transcription. Specifically, the authors proposed, based on their results, that PARP-1 modulates histone variant H2A.Z levels in the promoter proximal region through its PARylation activity. In their paper, the authors have integrated various aspects of published work to generate the hypotheses in their manuscript. From their results, the authors conclude that PARP-1 promotes dynamic remodeling of promoter-associated nucleosomes to allow transcriptional activation in response to cellular signaling.

Reviewer's comments:

This paper addresses an interesting and timely subject area that builds upon growing evidence in the literature for interactions between PARP-1 and H2A.Z (or the fly H2A.V ortholog). The

experiments presented are reasonably well done and the conclusions generally follow from the results. The major shortcomings of this paper are the lack of biochemical assays that could confirm and elucidate in more detail the proposed mechanisms, as well as experimental controls that are necessary to support the conclusions made. The latter is a serious issue that affects the overall interpretation of the results.

1) The authors absolutely need to include in their ChIP experiments a control for the total histone levels (i.e., an anti-H3 control). The authors describe a mechanism by which H2A.Z is removed and replaced with Histone H2A. The most significant evidence is the dramatic loss of H2A.Z at the locus. Yet, without accounting for total histone levels by including an anti-H3 ChIP, this conclusion cannot be supported. Moreover, showing that H2A occupancy increases in the H2A.Z loss experiments is another important control.

2) The authors have included an important experiment showing PARP-1^{-/-} primary cells exhibit the same molecular phenotype as other genetic and pharmacological treatments targeting PARP-1 and its activity. An important control that is lacking is re-expression of wt PARP-1 in the knockout background, showing that it can rescue this phenotype. Also, the authors should include re-expression of a catalytically impaired PARP-1 mutant, such as the classic E988K mutant, in order to demonstrate that the defects in FOS expression are mediated through the loss of PARP-1 derived PARylation.

3) The authors perform a series of knockdown experiments in order to parse out the molecular mechanisms occurring at the FOS gene locus. These conditions lack critical controls that need to be included in this manuscript. All of the knockdown experiments include a demonstration of knockdown at the mRNA level by RT-qPCR, but do not include a Western blot demonstration that the protein levels have also been knocked down. Moreover, in the conditions where ChIP is being performed after knockdown, control experiments showing that the RNAi-targeted proteins are actually lost from the locus are critical.

4) This paper lacks biochemical assays (e.g., *in vitro* reconstitutions) that can test in detail the proposed model. Can PARP-1 really promote the exchange of H2A.Z by H2A with purified nucleosomes? What is targeted by for PARylation?

5) The authors should provide the full anti-PAR Western blot in the inhibitor titration present in their supplemental figures including molecular weight markers. In our experience, PAR blots do not yield such discrete bands as shown by the authors.

6) The authors should include PARP-1 and PAR ChIP conditions in the experiments shown in Fig. 6, as it is unclear how the depletion of p400, INO80, and H2AZ affect PARP-1 activation and localization.

Referee #3

O'Donnell and Sharrocks provide evidence that NFI recruitment to the FOS promoter permits PARP1 binding, which is required for the expression of FOS. They further imply that in response to ERK pathway signaling, H2A.Z is replaced by H2A in a PARP1 and ADP-ribosylation-dependent manner. To support this model, the authors used both pharmacological and molecular approaches.

Although addressing an interesting concept, the provided data in this study do not fully support the conclusions by the authors. A few suggestions for improvement:

- The authors should further elucidate the molecular mechanism of PARP1 enzymatic activation, as ADP-ribosylation seems to be an important aspect. Is H2AZ activating PARP1 (see Kotova et al., 2011) or the acetylation status of NFI? Including e.g. a modification deficient NFI or H2AZ mutant would help to resolve these questions.

- Automodification of PARP1 was reported to lead to dissociation of PARP1 from the chromatin by different groups. It is thus very surprising that the authors detect PARP1 and PAR at the FOS

promoter! The author should provide evidence that the formed PAR is indeed synthesized by PARP1 (repeating the ChIP in the PARP1 knockout background).

- Moreover, in the discussion the authors speculate about a possible mechanism how PARP1 is regulating the dynamics of H2A.Z deposition through either affecting the activity of p400 and/or INO80, or by modifying directly H2A or H2A.Z. It would strengthen the manuscript if the authors would address these aspects in more detail (e.g., in vitro parylation assay using recombinant PARP1 and histone H2A and H2A.Z or p400 and INO80).

- The authors did not consider the possibility that PARP1 could affect H2A.Z dynamics indirectly by modifying H3K9 acetylation. PMA and PARP1-dependent H3K9ac might influence the chromatin association of H2A.Z. The authors could test this hypothesis by using an acetyltransferase inhibitor and analysing the H2A.Z replacement.

- Is the observed effect specific for ERK signaling or are other signaling cascades exhibiting the same regulatory mechanism (other H2AZ controlled genes)?

- The authors used different PARP1 inhibitors for separate experiments (3AB, KU00589948, Pj34), assuming that they have the same effect. Is this indeed the case?

- Fig.1 (all panels): The authors should include the IgG control in all panels and provide evidence that at an unrelated chromatin domain (e.g., -3000 from Fig. 4) is not recruitmented.

- Fig.2A/C: The authors should provide evidence that the protein levels (e.g., of p300, and especially of NFI) are not reduced in PARP1 knockout or siPARP1 treated cells, as these would explain why less NFI is recruited in these cells (Fig.3D).

- Fig.2C/3D: The reported reduced expression of FOS in PARP1 knockout cells should be controlled by genetically complementing these cells with wild type PARP1 and an enzymatically inactive mutant of PARP1. This should also be considered for Fig. 7C, as these are very important experiments to strengthen the authors' conclusions.

- Fig.3C/D: The authors performed NFI ChIP in siPARP1 treated cells and in PARP1 knockout cells at different time point (10 vs 20 min) without providing an explanation. It would be interesting to include the 20 min time point using PARP1 knockout cells.

- Fig.2E: The enhanced expression of FOS after 80 min does not support the authors' model and should thus be investigated in more detail (i.e., NFI and PARP1 recruitment is no longer observed). The analysis could be extended by performing ChIP for H2A and H2AZ at a later time points; 80 minutes.

- Fig.3F: The immunoprecipitation should be repeated vice versa (IP NFI and WB PARP1).

- Fig.4A: The authors observe a PARP1-dependent reduction of H3K9ac at the promoter of FOS. Repeating the experiments for the H3 occupancy for the same time points and promoter would allow the normalization of the data (H3K9ac/H3).

- Fig.4B: The authors should include an untreated sample and an unrelated chromatin domain (e.g., -3000) in their CHART-PCR analysis.

- Fig.5A: The IgG control is missing; panel C is mislabeled (replace p400 by INO80); Fig. 5D siRNA of p400 and INO80 should both be included in the presented ChIPs.

- Fig.6C: INO80 siRNA should be included to complete the analysis; Fig.6D include untreated samples and the analysis at an unrelated chromatin region.

- Fig.6G/7B: The analysis should be extended by measuring the H2A enrichment and by including more time points (as the CHART PCR is performed after 60 minutes).

- Fig.7B: In this panel, H2A.Z is released from the chromatin after 5 min of stimulation and recruited again after 10 min. According to the authors' model, this should repress FOS expression?! Moreover, this panel is inconsistent with data presented in Fig. 6G (no detectable H2A.Z enrichment after 10 min PMA stimulation).
- The immunoprecipitation should be repeated vice versa (IP PARP1 and WB p400 or INO80).
- Fig.S2A/B: Knockdown of PARP1 or PARP2 should be confirmed by WB.
- Fig.S3B: The reduction in FOS expression for shorter time points (40 min) cannot be appreciated. The authors should provide the same analysis also at 40 min time point.
- Fig.S4: WB for PARP1 should be included.

Additional correspondence (author)

21 December 2012

Thanks for the decision on this paper. We have decided to attempt these revisions as we believe our story warrants publication in a high impact journal like EMBO J. There will be a delay as the Postdoc who did all the work is on maternity leave for another 11 months and we need to fill in with others. I think we can address everything as the issues raised are mainly controls rather than anything that will change our story. One issue I want to clarify is your statement that "the model you propose should be tested in in-vitro reconstitution assays". What we plan to do is test one of the things that the referee suggests ie that histones (presumably H2AZ) are targets (or not) for parylation. This is the simplest model. What we cannot do is reconstitute the whole system, as this is just technically infeasible, and would actually constitute a paper in its own right (and probably another 2-3 years work). Such a system would involve reconstituting the FOS promoter in its correct chromatin context (which involves unknown stoichiometry of H2AZ/H2A, which we somehow need to localise only at one position on the proximal nucleosome), then recombinant (maybe modified), ELK1, SRF, p300, NFI, PARP, INO80 (a complex) and P400 (a complex), and then see whether this system is sufficient to work in response to ELK1 phosphorylation (which I doubt as we will be missing many factors). Clearly this is not feasible.

Can you therefore let us know either our plan is reasonable, and we will of course qualify our model accordingly, making it clear what is a possibility, and what we have ruled in or ruled out. In the end, our key conclusion is that PARP1 orchestrates a lot of dynamic changes to the promoter of which histone exchange is just one of the prominent ones we wished to highlight. We will provide more robust data to further demonstrate this key finding.

I look forward to hearing your response.

Additional correspondence (editor)

22 December 2012

Thank you for your comments in response to my decision. I am happy to hear that you decided to embark on the requested revisions. Please just contact me in case you will not be able to meet our usual three month timeline. If the conceptual advance of the study is not eroded by publications that have appeared in the meantime, we can certainly grant an extension of the revision period.

I definitely understand that it will not be feasible to reconstitute your whole experimental system in vitro. With my statement, I was referring specifically to Point 4) of Ref #2 and Point 3 of Ref # 3. I hope this clarifies our experimental demands in that regard. I would also like to mention at this point that acceptance of the manuscript is likely to entail validation of your revision with a subset of the referees.

I am looking forward to your revision!

Issues raised by the Editor

You raised four issues that we needed to address in your summary statement:

- (1) “*However, they raise significant concerns, especially the lack of pertinent controls.....*”. We have now added numerous controls as requested (see details below).
- (2) “*In particular, kinetics of the observed phenomena need to be more firmly established....*”. We appreciate the problems with inconsistent kinetics seen in our initial paper (see below for more detailed discussion of this issue). We have therefore re-done several experiments and focussed on the *Parp1* knockout cells to enable us to match activation kinetics with ChIP events more precisely by doing the experiments in parallel.
- (3) “*A rescue experiment of PARP1 -/- cells with either wild-type or catalytically inactive PARP should be included.....*” This is an important suggestion, and we have now done this key experiment. The results are fully consistent with a role for catalytically active PARP1 in controlling *FOS* activation and have solidified the molecular mechanism we have proposed.
- (4) “*The model you propose should be tested in in-vitro reconstitution assays*”. The simplest model that we proposed was that PARP1 causes differential parylation of histones H2AZ and H2A, and hence contributes to their exchange on the *FOS* promoter. We have tested this, and found evidence that H2AZ is parylated *in vitro* but there was little evidence for H2A parylation. However, given this result we realised that it would then require a whole new study to demonstrate the functional consequences of H2AZ parylation *in vivo*. Indeed, more complex scenarios are likely involved which is hinted at by our observation of PARP-dependent changes in p400 occupancy (new mechanistic data now included in the paper). To properly address the molecular complexities, we would therefore need to perform an extensive new study and hence we believe this is outside the scope of the already complex and detailed present study.

In summary, we believe that the paper is now much better controlled and now has extensive evidence to support the major claim of our paper ie that PARP1 activity is needed for signal-dependent variant histone exchange and hence gene activation.

Referee 1

This reviewer raises a number of specific issues to address:

- (1) *In figure 3E, the authors use a PARP1 inhibitor, PJ34, to show the function of PARP1 activity for NFI recruitment. Why do the authors use this particular inhibitor here, since two different inhibitors are used in figure 2F? The effect of 3AB and KU0058948 on NFI recruitment should be shown. The effect of PJ34 on FOS transcription should also be shown. If particular inhibitors are not used, it should at least be justified.*

Response: We commonly test at least two different inhibitors in each assay and have actually used both inhibitors in this case but chose to only show one of them. We have now added the data using the 3AB PARP1 inhibitor, and as observed with PJ34, treatment with 3AB also reduces NFI recruitment (data now included in Supplementary Fig. S5).

- (2) *In figure 5D, the authors show that a knock-down of p400 decreases H2A.Z and a knock-down of Ino80 decreases H2A at the FOS promoter. p400 has been shown to catalyze the exchange of H2A for H2A.Z, so it would be expected that knock-down should decrease H2A.Z with an increase in H2A. The authors should show a ChIP for H2A in a p400 knock-down and a ChIP for H2A.Z in a Ino80 knock-down.*

Response: We have done the required experiment. However, while knockdown of p400 specifically reduces H2A.Z levels and knockdown of INO80 reduces H2A levels as expected, there is no reciprocal increase in the levels of the H2A and H2AZ respectively at the promoter (see new Fig. 5E). This suggests that the effects we see are not simply the result of disturbing a dynamic equilibrium, but represent regulated p400- and INO80-dependent events. Ie loss of either exchange

factor will cause loss of a particular H2A isoform at the promoter, but this does not automatically lead to more deposition of an alternative replacement isoform. This finding is now discussed in the revised manuscript.

(3) *The authors show that both p400 and Ino80 are present at the FOS promoter (Figure 5B,C). What are the kinetics of p400 and Ino80 after PMA treatment? This is important, since it could explain the kinetics of H2A.Z observed at the promoter.*

Then authors then show that PARP1 interacts in vivo with p400 and Ino80 proteins (Figure 7A). An obvious model is that PARP1 or its catalytic activity could function in the recruitment of p400 and Ino80 to the promoter. The authors should perform p400 and Ino80 ChIPs in WT and Parp -/- cells (or p400 or Ino80 siRNA knock-down cells) or cells treated with 3AB after PMA treatment. This experiment is well within grasp of the authors.

Response: We have now examined the binding kinetics of p400 and INO80 at the *FOS* promoter in wild-type and *parp1* knockout MEFs. We find binding of both enzymes throughout the timecourse of PMA treatment. However, we do see dynamic changes in the levels of both p400 and INO80 at the *FOS* promoter in wild-type cells, and these changes are abolished in *parp1* KO cells, demonstrating an important role for PARP1 in controlling these dynamics (new Fig. 5D). Importantly, increased parylation levels at the *FOS* promoter occur as p400 levels are reduced, underlining the role of PARP1 in controlling this binding activity. Importantly, decreased levels of p400 binding precede reductions in H2AZ incorporation in MEFs (see new Fig. 7B) consistent with a role for this enzyme being required for H2AZ deposition. Increases in INO80 binding later in the timecourse likely reflect the requirement to redress the balance between p400 and INO80 activities as p400 binding increases again. These issues are now discussed further in the revised manuscript.

(4) *All through the manuscript, the time points taken after PMA treatment for ChIP experiments are inconsistent and no explanation is given. In experiments where a single time point is taken, cells were harvested after either 5min (Figure 6D) or 10 min (other Figures) after PMA treatment. In other experiments, multiple time points were taken at either 5 and 20 min (Figure 1B and 3C), 3 and 10 min (Figure 3B), 5 and 10 min (Figure 3D). This is not trivial, since the dynamics of protein binding to the FOS promoter are very fast. The authors should at least explain these discrepancies.*

Response: The reviewer makes an important point here. We have added further experiments to address these concerns (see below). The reason for the different kinetics likely stems partly from the fact that experiments were done over a long time period (4-5 years) and over that time growth conditions, reagents and cells likely change slightly which would give rise to small temporal changes in the events we see. Furthermore, there is also the issue of different cell types ie human HeLa cells versus mouse MEFs, whose wiring may not be identical. We have now added further explanation where necessary for why certain timepoints were taken where this is not clear. Importantly, we now have repeated critical timecourse ChIP data (see below).

(4 cont.....) *This point is particularly important in Figure 7B, where H2A.Z is shown to be evicted from the promoter at 5 min, and then returns as soon as 10 min. Then, in Figure 7C, H2A.Z ChIPs are shown for a single time point at 10 min only. Wouldn't it have been more pertinent to perform the ChIP at 5 min, when maximal H2A.Z eviction is expected to be achieved? Given the rapid kinetics, the authors should perform H2A.Z ChIPs at multiple time points after PMA treatment in WT and Parp -/- cells to verify the effect of PARP1 on H2A.Z dynamics.*

Response: The referee makes an important point but here we are comparing HeLa cells (Fig. 7B; and MEFs (Fig. 7C). We agree that the different kinetics is confusing, therefore we have moved Fig. 7B and 7C to the supplementary data (Fig. S10), and replaced this with new timecourse data that show temporal H2AZ enrichment levels at the *FOS* promoter in wild-type and *parp1* deficient MEFs. This new data agree with our previous results which show transiently reduced H2AZ levels at the *FOS* promoter in WT MEFs, but little change in *parp1* deficient MEFs.

(5) *In Figure S2B, is there a control missing? As I understand the figure, it seems that the effect of PARP2 knock-down on GAPDH is missing.*

Response: The point of this figure is to demonstrate that PARP2 siRNAs lead to PARP2 depletion, and that PARP1 siRNAs lead to PARP1 depletion. This is clearly shown. We have now added a western blot to show that these effects seen at the mRNA level are also reflected at the protein level (Supplementary Fig. S2C).

(6) *In Figure S5, the authors perform mono-nucleosome ChIP to verify that the -1 nucleosome is enriched in H2A.Z. Since the cells were fixed with formaldehyde, it is possible that contaminating di-nucleosomes are in the extract and that neighbouring nucleosomes are contributing to the H2A.Z signal. The authors should provide control experiments that were done to ensure minimal di-nucleosome contamination. Also, they could easily pinpoint the location of H2A.Z using PCR probes spanning this region (as in Figure 6G) on the same samples.*

Response: As requested by the referee, we now provide evidence that our chromatin preparation in this experiment consisted predominantly of mononucleosomes (new Supplementary Fig. S7C).

Minor comments:

(7) *In Figures 4B the CHART-PCR normalization is not clear. How is % access to DNA actually calculated? It is said that 100% is non-digested input DNA. So in that case is access simply (1-% of amplification)? It should be explained in the methods section.*

Response: We have amended the materials and methods section to explain this more clearly.

(8) *In p.10, line 10. The sentence starting with "Since NFI and PARP1 associate with a highlypositioned..." reads poorly and needs to be revised.*

Response: This has been changed to make this clearer.

(9) *In Figure 5C, I think the axis should say INO80 instead of p400.*

Response: This error has now been corrected.

(10) *In p12, 3rd paragraph. The sentence starting with "Having established that PARP1 is associated.." reads poorly and needs to be revised.*

Response: This has been changed to make this clearer.

Referee 2

Several specific issues are raised by this referee:

(1) *The authors absolutely need to include in their ChIP experiments a control for the total histone levels (i.e., an anti-H3 control). The authors describe a mechanism by which H2A.Z is removed and replaced with Histone H2A. The most significant evidence is the dramatic loss of H2A.Z at the locus. Yet, without accounting for total histone levels by including an anti-H3 ChIP, this conclusion cannot be supported. Moreover, showing that H2A occupancy increases in the H2A.Z loss experiments is another important control.*

Response: The data in Fig.5 are already normalised for histone H3 levels so this has already been taken into account. However, the referee makes an important point which we have now addressed in the context of other experiments:

First, we have investigated H3 levels in MEFs following PMA stimulation. In wild-type MEFs, we see no large decrease in H3 levels after 10 minutes which would explain the loss of H2AZ signal (new Supplementary Fig. S9C). At the same timepoint there is an increase in H2A levels.

Secondly, we have now shown the deconvoluted data from Fig. 6G in Supplementary Fig. S9. This clearly shows that H2AZ levels at the *FOS* promoter decrease after PMA stimulation whereas H2A levels increase. Thus H2AZ loss is accompanied by H2A gain. We have also added in an additional timepoint into Fig. 6G, so the gradual time-dependent diminution of H2AZ signal is clearly shown.

(2) *The authors have included an important experiment showing PARP-1^{-/-} primary cells exhibit the same molecular phenotype as other genetic and pharmacological treatments targeting PARP-1 and its activity. An important control that is lacking is re-expression of wt PARP-1 in the knockout background, showing that it can rescue this phenotype. Also, the authors should include re-expression of a catalytically impaired PARP-1 mutant, such as the classic E988K mutant, in order to demonstrate that the defects in FOS expression are mediated through the loss of PARP-1 derived PARylation.*

Response: This is an excellent suggestion of an experiment, which we have now performed. The results are now shown in Figs. 2G and H. Wild-type PARP1 is able to restore increased peak PMA responsiveness to the *FOS* locus in *parp1* KO MEFs but the catalytically dead E988K version is unable to do so.

(3) *The authors perform a series of knockdown experiments in order to parse out the molecular mechanisms occurring at the FOS gene locus. These conditions lack critical controls that need to be included in this manuscript. All of the knockdown experiments include a demonstration of knockdown at the mRNA level by RT-qPCR, but do not include a Western blot demonstration that the protein levels have also been knocked down. Moreover, in the conditions where ChIP is being performed after knockdown, control experiments showing that the RNAi-targeted proteins are actually lost from the locus are critical.*

Response: We generally check for protein knockdown at the start of experiments to validate each siRNA that we use. However, once validated, we then use RNA levels as a proxy for protein loss. As requested we have now added some westerns to show loss of protein. For example, we have now added a western blot to show that PARP2 siRNAs lead to PARP2 depletion, and that PARP1 siRNAs lead to PARP1 depletion (New Fig. S2C). We have also added western blots for H2A.Z, p400 and INO80 (new supplementary Fig. S8D-F) to show that the respective proteins are lost upon knockdown.

(4) *This paper lacks biochemical assays (e.g., in vitro reconstitutions) that can test in detail the proposed model. Can PARP-1 really promote the exchange of H2A.Z by H2A with purified nucleosomes? What is targeted by for PARylation?*

Response: The simplest model we proposed was that PARP1 directly parylates H2A and/or H2AZ and thereby directly contributes to their binding to the promoter-proximal nucleosome. We began to test this hypothesis and found that while H2A.Z is PARylated *in vitro*, we can find little evidence for H2A parylation. However, to incorporate this result into a detailed model, we realised that this would take extensive experimentation that would constitute an extensive completely new study. Indeed the mechanism through which PARP1 is functioning is likely to be more complicated. For example, p400 or a co-binding protein might be parylated which might change their activity towards histone deposition. Indeed, in the current study we now include new data to show that p400 binding is dynamically altered during the promoter activation process, and that this event is dependent on PARP activity. However, testing such things in reconstituted biochemical systems is well beyond the scope of the current study. We have amended our discussion section accordingly to reflect our preliminary findings.

(5) *The authors should provide the full anti-PAR Western blot in the inhibitor titration present in their supplemental figures including molecular weight markers. In our experience, PAR blots do not yield such discrete bands as shown by the authors.*

Response: We have now included an extended version of the western blot as requested. A more extended ladder of parylation is now seen, but with a single major band which corresponds to parylated PARP1. In the previous version, the blot had become inadvertently compressed, obscuring the broader spread of parylated species.

(6) *The authors should include PARP-1 and PAR ChIP conditions in the experiments shown in Fig. 6, as it is unclear how the depletion of p400, INO80, and H2AZ affect PARP-1 activation and localization.*

Response: The referee asks whether we see reciprocal effects on PARP1 binding following depletion of H2AZ and its regulators. To address this, we depleted H2AZ and assayed PARP1 binding but were unable to see substantial changes in PARP1 binding to the *FOS* promoter. This suggests that PARP1 acts upstream from H2AZ in this regulatory mechanism. We have mentioned this in the revised manuscript as “data not shown”.

Referee 3

This referee provides several suggestions for improvement:

(1) *The authors should further elucidate the molecular mechanism of PARP1 enzymatic activation, as ADP-ribosylation seems to be an important aspect. Is H2AZ activating PARP1 (see Kotova et al., 2011) or the acetylation status of NFI? Including e.g. a modification deficient NFI or H2AZ mutant would help to resolve these questions.*

Response: The purpose of the current study is to establish the involvement and role of PARP1 in *FOS* promoter activation. The referee poses an interesting question based on a recent publication but this is not really within the scope of the current paper and would be the subject of a differently designed study with alternative goals. We do however now allude to negative data that we have which shows that H2A.Z depletion does not greatly affect PARP1 recruitment. This suggests that genetically, H2A.Z is downstream from PARP1, at least in terms of assembly events on the promoter region.

(2) *Automodification of PARP1 was reported to lead to dissociation of PARP1 from the chromatin by different groups. It is thus very surprising that the authors detect PARP1 and PAR at the FOS promoter! The author should provide evidence that the formed PAR is indeed synthesized by PARP1 (repeating the ChIP in the PARP1 knockout background).*

Response: We have now performed the requested experiment and find that the PMA-mediated increase in PAR signal at the *FOS* promoter is abolished in *parp1* knockout MEFs (see new Fig. 5D).

(3) *Moreover, in the discussion the authors speculate about a possible mechanism how PARP1 is regulating the dynamics of H2A.Z deposition through either affecting the activity of p400 and/or INO80, or by modifying directly H2A or H2A.Z. It would strengthen the manuscript if the authors would address these aspects in more detail (e.g., in vitro parylation assay using recombinant PARP1 and histone H2A and H2A.Z or p400 and INO80).*

Response: The simplest model we proposed was that PARP1 directly parylates H2A and/or H2AZ and thereby directly contributes to their binding to the promoter-proximal nucleosome. We began to test this hypothesis and found that while H2A.Z is PARylated *in vitro*, we can find little evidence for H2A parylation. However, to incorporate this result into a detailed model, we realised that this would take extensive experimentation that would constitute an extensive completely new study. Indeed the mechanism through which PARP1 is functioning is likely to be more complicated, and there may be many targets of for PARYlation at the *FOS* promoter. For example, p400 or a co-binding protein might be parylated which might change their activity towards histone deposition. Indeed, in the current study we now include new data to show that p400 binding is dynamically altered during the promoter activation process, and that this event is dependent on PARP activity. However, testing such things in reconstituted biochemical systems is well beyond the scope of the current study. We have amended our discussion section accordingly to reflect our preliminary findings.

(4) *The authors did not consider the possibility that PARP1 could affect H2A.Z dynamics indirectly by modifying H3K9 acetylation. PMA and PARP1-dependent H3K9ac might influence the chromatin association of H2A.Z. The authors could test this hypothesis by using an acetyltransferase inhibitor and analysing the H2A.Z replacement.*

Response: This is an interesting possibility but histone acetylation is required for NFI recruitment (O'Donnell et al., 2008) and therefore as shown in the current paper, the subsequent recruitment of PARP1 (which is NFI dependent). Thus inhibition of histone acetylation would stop the initiation of the whole process, and hence it would be impossible to study the effects on PARP1-dependent H2AZ replacement. We have therefore not attempted this experiment.

(5) *Is the observed effect specific for ERK signaling or are other signaling cascades exhibiting the same regulatory mechanism (other H2AZ controlled genes)?*

Response: A previous study demonstrated that doxorubicin treatment leads to dynamic changes in H2AZ deposition at the *p21* promoter (Gevry et al., 2007). We therefore tested whether PARP1 was required for this effect, by comparing H2AZ levels at the *p21* promoter in wild-type and *parp1* knockout MEFs. While H2AZ enrichment levels decreased after 5 hrs doxorubicin treatment, no such decrease was observed in *parp1* KO MEFs. Therefore PARP1 is required for signal induced H2AZ loss in response to a different inducer at a different promoter region, albeit with different kinetics.

(6) *The authors used different PARP1 inhibitors for separate experiments (3AB, KU00589948, Pj34), assuming that they have the same effect. Is this indeed the case?*

Response: We have compared the effects of at least two and sometimes three different PARP1 inhibitors and see little difference in the molecular outcomes. We have now added an additional experiment to illustrate this where we examine NFI recruitment with an alternative inhibitor, 3AB and find that this gives an identical effect to the PJ34 inhibitor and reduces NFI recruitment (data now included in Fig. 1E and Supplementary Fig. S5).

(7) *Fig.1 (all panels): The authors should include the IgG control in all panels and provide evidence that at an unrelated chromatin domain (e.g., -3000 from Fig. 4) is not recruited.*

Response: We routinely check all antibodies upon first use for enrichment above IgG and against a negative control locus. For simplicity, we chose not to include these in the original manuscript but now add the fully controlled figures for PARP1 and PAR CHIP to Fig. 1 (new panels Fig. 1A and D) to demonstrate the specificity of the signals we observe.

(8) *Fig.2A/C: The authors should provide evidence that the protein levels (e.g., of p300, and especially of NFI) are not reduced in PARP1 knockout or siPARP1 treated cells, as these would explain why less NFI is recruited in these cells (Fig.3D).*

Response: We checked p300 and NFI levels in wild-type and *parp1* knockout MEFs. Importantly no decrease in their levels was seen in the KO MEFs and increased levels were actually observed (especially of p300)(see Fig. S2D). Thus PARP1 loss does not lead to reductions in the levels of either of these regulators, ruling out a trivial explanation as to why less NFI is recruited.

(9) *Fig.2C/3D: The reported reduced expression of FOS in PARP1 knockout cells should be controlled by genetically complementing these cells with wild type PARP1 and an enzymatically inactive mutant of PARP1. This should also be considered for Fig. 7C, as these are very important experiments to strengthen the authors' conclusions.*

Response: This is an excellent suggestion of an experiment, which we have now performed. The results are now shown in Figs. 2G and H. Wild-type PARP1 is able to restore increased peak PMA responsiveness to the *FOS* locus in *parp1* KO MEFs but the catalytically dead E988K version is unable to do so.

(10) *Fig.3C/D: The authors performed NFI CHIP in siPARP1 treated cells and in PARP1 knockout cells at different time point (10 vs 20 min) without providing an explanation. It would be interesting to include the 20 min time point using PARP1 knockout cells.*

Response: We performed the experiments in HeLa cells at time points where we had already established the rapid and transient NFI recruitment kinetics (O'Donnell et al., 2008). Having established the role of PARP1 in the transient recruitment, we examined this phase of the activation

process in MEFs, hence why we did not take additional timepoints beyond this. The results are internally consistent ie NFI recruitment is reduced upon depletion of PARP1 in two different cell types using different strategies, and the experiments are sufficient to make this point without including additional timepoints. The rationale for the experiments has now been incorporated into the text. Overall, the key thing here is to illustrate the reductions in NFI recruitment in the two systems, and that is clearly illustrated by the experiments shown.

(11) *Fig.2E: The enhanced expression of FOS after 80 min does not support the authors' model and should thus be investigated in more detail (i.e., NFI and PARP1 recruitment is no longer observed). The analysis could be extended by performing ChIP for H2A and H2AZ at a later time points; 80 minutes.*

Response: The referee makes a good point, but we are not focussing on the later timepoints in the context of this paper, but instead are looking at the early activating events. We have data showing that inhibition of PARP1 leads to hyperinduction of coactivator binding and loss of corepressor binding at later timepoints. However, further characterisation of this interesting phenomenon will constitute a separate study.

(12) *Fig.3F: The immunoprecipitation should be repeated vice versa (IP NFI and WB PARP1).*

Response: We understand the referee's point but as the result is convincing as it stands, we do not believe that reversing the co-IP is necessary.

(13) *Fig.4A: The authors observe a PARP1-dependent reduction of H3K9ac at the promoter of FOS. Repeating the experiments for the H3 occupancy for the same time points and promoter would allow the normalization of the data (H3K9ac/H3).*

Response: Fig. 4A is already normalised for total H3 levels (as described in the legend)

(14) *Fig.4B: The authors should include an untreated sample and an unrelated chromatin domain (e.g., -3000) in their CHART-PCR analysis.*

Response: New control data has now been added to the manuscript (see Fig. S6A and B) that shows that PMA specifically affects nucleosome accessibility at the "-1" nucleosome. As the accessibility does not change at the "-3000" region there can be no change in response to PMA whether or not PARP1 is there or not.

(15) *Fig.5A: The IgG control is missing; panel C is mislabeled (replace p400 by INO80); Fig. 5D siRNA of p400 and INO80 should both be included in the presented ChIPs.*

Response: For Fig.5A, the data are presented as relative to histone H3, therefore control IgG was not shown. However, we have now added additional supplementary data to show that the ChIP signal for H2A.Z is significantly higher than for control IgG (new Supplementary Fig.S7A). The efficiency of siRNA knockdown is already shown in Supplementary Fig. S8.

(16) *Fig.6C: INO80 siRNA should be included to complete the analysis; Fig.6D include untreated samples and the analysis at an unrelated chromatin region.*

Response: We inadvertently omitted the INO80 data and this has now been added to Fig. 6C. We have now added an untreated control sample to accompany Fig. 6D and this is shown in Fig. S8G.

(17) *Fig.6G/7B: The analysis should be extended by measuring the H2A enrichment and by including more time points (as the CHART PCR is performed after 60 minutes).*

Response: The data in these two figures show a ratio of H2AZ to H2A as stated in the figure legends, thereby providing H2A.Z enrichment levels. H2A enrichment levels would therefore just be the reciprocal of this. We appreciate the point that the referee is making. However, the two assays are different. In Fig. 6C, we wished to demonstrate that chromatin accessibility is increased at the FOS promoter upon depletion of either H2A.Z or p400. As such the time point is largely irrelevant, as H2AZ will be lost, and hence the promoter more open and accessible irrespective of the time of

PMA stimulation. We previously showed that during PMA induction, the promoter becomes more accessible 10 mins after PMA stimulation, which is consistent with the times seen for changes in H2AZ occupancy in Fig. 6G (an additional timepoint now added to help illustrate this point better). We have now referred back to this previous result when discussing the CHART PCR result in Fig. 4 to clarify things in the revised paper.

(18) *Fig.7B: In this panel, H2A.Z is released from the chromatin after 5 min of stimulation and recruited again after 10 min. According to the authors' model, this should repress FOS expression?! Moreover, this panel is inconsistent with data presented in Fig. 6G (no detectable H2A.Z enrichment after 10 min PMA stimulation).*

Response: The referee makes an important point about inconsistent kinetics. There are many technical reasons why this might be the case. To avoid confusion, we have therefore removed the original Fig. 7B and replaced this with new timecourse data that show temporal H2AZ enrichment levels at the *FOS* promoter in wild-type and *parp1* deficient MEFs. This new data agree with our previous results which show transiently reduced H2AZ levels at the *FOS* promoter in WT MEFs, but little change in *parp1* deficient MEFs.

(19) *The immunoprecipitation should be repeated vice versa (IP PARP1 and WB p400 or INO80).*

Response: We understand the referee's point but as the result is convincing as it stands, we do not believe that reversing the co-IP is necessary.

(20) *Fig.S2A/B: Knockdown of PARP1 or PARP2 should be confirmed by WB.*

Response: This has now been added to Supplementary Fig.S2C.

(21) *Fig.S3B: The reduction in FOS expression for shorter time points (40 min) cannot be appreciated. The authors should provide the same analysis also at 40 min time point.*

Response: We agree that looking at FOS protein expression at earlier timepoints might reveal even earlier induction by 3AB treatment. However, the point of this experiment is to show that the increased mRNA expression levels seen after 60 mins (Fig. 2E) result in higher levels of protein. This is illustrated in the current figure, therefore we have chosen not to repeat the experiment with additional earlier timepoints. The text has been amended slightly to explain the rationale for the experiment more clearly.

(22) *Fig.S4: WB for PARP1 should be included.*

Response: A western blot has now been included as requested.

2nd Editorial Decision

03 July 2013

Thank you for submitting your revised manuscript for our consideration. It has now been seen once more by the original referees, whose comments are provided below.

Although the reviewers acknowledge that the revision has improved the manuscript, referee 2 and 3 still find that your study currently falls short of providing the kind of molecular insight into how PARP-1 affects histone variant exchange that would be expected from an EMBO Journal paper. While referee 1 is more positive in this regard, your ability to extend the mechanistic depth of your manuscript was an important criterion in our initial editorial decision. We were aware that this demanded challenging and time-consuming experiments, a fact that I stressed in my original decision letter.

Given our policy to allow only a single major round of revision and the fact that two experts in the field are unable to support publication of your manuscript in its current form in The EMBO Journal, I am afraid that we cannot offer further consideration.

I appreciate that you invested significant time and effort in this revision, and am very sorry that I cannot be more positive on this occasion. I hope that you nevertheless found the referees' comments helpful and that the implemented changes will ease publication elsewhere.

REFEREE COMMENTS

Referee #1

I read the rebuttal and new manuscript. I have also taken into account their new experiments/controls. I have to say I am quite happy with their revisions. I think all reviewers asked challenging new experiments in some cases, which may not have been entirely realistic or fair to the authors. In sum, this is an important study that I think deserves to be published in the EMBO Journal.

Referee #2

In the revised version of their paper, the authors have addressed a number of significant concerns that were identified in the first round of review. These include the addition of important controls (e.g., normalizing for histone occupancy, re-expression of wt PARP-1 in the knockout background, Western confirmation of knockdown, etc.), which strengthen the experiments and conclusions to which they pertain. In addition, the authors have added an extended panel of experiments examining the role of p400 and INO80 in the activation of the FOS gene.

Having said that, a major concern from the previous round of review still remains, namely the lack of specific mechanistic links between PARP-1, p400/INO80, and H2AZ exchange. Fig. 5 establishes that (1) H2AZ, p400, and INO80 are enriched at the FOS promoter prior to induction, (2) H2AZ and p400 levels are rapidly reduced upon PMA treatment in a manner that is dependent on PARP-1 protein (based on Parp1 knockout MEFs), and (3) p400 is required for H2AZ occupancy, whereas INO80 is required for H2A occupancy. What these studies fail to do is connect PARP-1 enzymatic activity to a specific molecular outcome related to p400 or INO80.

In their discussion (p. 17), the authors conclude that PARP-1 is required for histone variant exchange - a very interesting and provocative conclusion, indeed. But, at the same time, they note that "it is not entirely clear how it achieves this."

The authors have talked themselves out of biochemical experiments to address histone exchange based on the added complexity of roles for p400 and INO80, after abandoning a more simplified model paper based on histone PARylation. This may be a reasonable strategy, but it still leaves the authors without a definitive mechanism. Is PARP-1 enzymatic activity required for the activity of p400 or INO80? Are p400 or INO80 PARylated by PARP-1? These are basic questions that could be addressed without nucleosome reconstitutions.

So, as before, I feel that this paper addresses an interesting and timely subject area that builds upon growing evidence in the literature for interactions between PARP-1 and H2AZ. But a major shortcoming remains, namely the lack of data that could confirm and elucidate in more detail the proposed mechanisms.

Referee #3

O'Donnell and Sharrocks provide a revised version of their manuscript (EMBOJ-2012-83512). Although the authors included several controls and addressed some of the criticized points, the new data do not provide additional mechanistic insights.

- The authors do not provide additional data regarding the molecular mechanism of PARP1 enzymatic activation, although ADP-ribosylation seems to be an important aspect.

- The authors indicate that H2AZ, but not H2A, is ADP-ribosylated in vitro. Unfortunately, they are not following up this observation to provide more insight.
- The newly provided experiments on the p21 promoter (Fig. S10D) are rather preliminary and should be further investigated (regarding the contribution of p400/INO80).
- This reviewer feels that IgG should be included in all figures (not only in Fig. 1), because it would strengthen the authors' experiments.
- The genetic complementation of PARP1 KO cells with PARP1 WT or with an inactive mutant support the authors' conclusion. It is however very surprising that they are not providing ChIP results for the complemented PARP1 WT and the inactive PARP1, since this would provide additional information as to which extent the enzymatic activity is required.
- Although the different time points are explained (Fig. 3C/D), including the missing time points would have clarified the results.
- Why are the authors still including the 80 min time point in Fig. 2E, if they declare to focus on early activating events?

In contrast to the authors' response, the immunoprecipitation in Fig. 3F/7B should be repeated vice versa (IP NFI and WB PARP1).

Although the authors state that they removed Fig. 7B, it reappears as Fig. S10C! Again, in this panel, H2A.Z is released from the chromatin after 5 min of stimulation and recruited again after 10 min. According to the authors' model, this should repress FOS expression?! Moreover, this panel is inconsistent with data presented in Fig. 6G (no detectable H2A.Z enrichment after 10 min PMA stimulation).

The knockdown control of PARP2 by WB is not at all convincing (Fig. S2C)

Appeal

04 July 2013

Thanks for this decision. Can I however ask whether sending this to an independent new referee is at all possible? I can send a full justification of why I think this is required if this is a viable route. I basically have to take issue with the fact that we have not "extended the mechanistic depth" as we have done so. The question can always be whether this is far enough or not but we believe that a fresh pair of eyes seeing the paper in its current form might be able to comment better.

If you would be willing to send this to a new reviewer, I would be happy to set out the reasons in more detail, as I think the referees are not being entirely reasonable in their requests at this stage (as indicated clearly by the first reviewer). There is little issue with the conclusiveness of our paper.....it is all about where the study should end.

Response to referee #2 (EMBOJ-2012-83512R)

In the revised version of their paper, the authors have addressed a number of significant concerns that were identified in the first round of review. These include the addition of important controls (e.g., normalizing for histone occupancy, re-expression of wt PARP-1 in the knockout background, Western confirmation of knockdown, etc.), which strengthen the experiments and conclusions to which they pertain. In addition, the authors have added an extended panel of experiments examining the role of p400 and INO80 in the activation of the FOS gene.

Response: We are glad that the referee finds our paper to be improved. We are also glad that he/she has recognised the extra effort that we have made to extend our analysis of p400 and INO80 involvement, and hence provide more mechanistic insight.

Having said that, a major concern from the previous round of review still remains, namely the lack of specific mechanistic links between PARP-1, p400/INO80, and H2AZ exchange. Fig. 5 establishes

that (1) H2AZ, p400, and INO80 are enriched at the FOS promoter prior to induction, (2) H2AZ and p400 levels are rapidly reduced upon PMA treatment in a manner that is dependent on PARP-1 protein (based on Parp1 knockout MEFs), and (3) p400 is required for H2A.Z occupancy, whereas INO80 is required for H2A occupancy. What these studies fail to do is connect PARP-1 enzymatic activity to a specific molecular outcome related to p400 or INO80.

Response: These series of connections between PARP-1 activity and H2A.Z exchange/exchange factors are novel connections. Importantly, we now show that PARP-1 is involved in the rapid transient loss of p400 at the promoter, which helps explain the loss of H2A.Z binding. We believe that this helps provide a mechanistic understanding of how PARP-1 affects histone exchange. Our new data show that PARP-1 is clearly involved in this process. If required we could repeat the experiment using a PARP inhibitor to provide direct evidence that it is the enzymatic activity of PARP-1 that is important in this context, as we have not formally made this connection.

In their discussion (p. 17), the authors conclude that PARP-1 is required for histone variant exchange - a very interesting and provocative conclusion, indeed. But, at the same time, they note that "it is not entirely clear how it achieves this."

Response: We are glad that the referee finds this to be a "very interesting and provocative conclusion". We think that this is key here, and the reason why we sent this to EMBO J. Our data clearly support this conclusion, and hence should stimulate others in the field to change their thinking. We agree that we do not have the complete mechanism but as explained before, a complete mechanism will take an immense amount of work. That said, we can provide some simple experimentation to move things on more (see below) and provide a strong hint at the mechanism involved.

The authors have talked themselves out of biochemical experiments to address histone exchange based on the added complexity of roles for p400 and INO80, after abandoning a more simplified model paper based on histone PARylation. This may be a reasonable strategy, but it still leaves the authors without a definitive mechanism. Is PARP-1 enzymatic activity required for the activity of p400 or INO80? Are p400 or INO80 PARylated by PARP-1? These are basic questions that could be addressed without nucleosome reconstitutions.

Response: We are glad that the referee recognises that performing in vitro reconstitution experiments to monitor histone exchange is too much of an undertaking to ask for in the context of this paper. This is of course the best way biochemically to tackle this problem, and ultimately, the definitive proof required. Proof would also require in vivo demonstration of the same phenomenon. I.e mapping and mutation of the parylation site(s) on the target protein(s), replacement of endogenous alleles, and then monitoring the chromatin environment of the FOS promoter. It is for these reasons that we decided not to attempt to solve this mechanism in the context of this paper, as it is difficult to prove anything with a whole suite of experiments.

The referee suggests two possible ways forward:

- (1) *Is PARP-1 enzymatic activity required for the activity of p400 or INO80?* In theory this could be addressed in vitro but it is possibly (and indeed likely) that there might be a promoter-specific effect in vivo. Our new data suggest that PARP-1 is required for controlling dynamic binding of these enzymes, and therefore such a mechanism would not be required (although clearly could contribute). The experiment indicated above would address whether PARP-1 enzymatic activity is needed for controlling temporal p400/INO80 binding events.
- (2) *Are p400 or INO80 PARylated by PARP-1?* We could address this possibility although this would first require the purification of multi-component protein complexes as these are just two components of such complexes. This in itself is a lot of work, and then after identifying the protein(s) parylated in vitro, we would need to confirm this in vivo, after site mapping, mutation and allele exchange, followed by functional studies. Again, this is an immense amount of work.

We do however have another suggestion which will determine whether a PARP-1 target in vitro is potentially involved in vivo. We have shown that H2A.Z is parylated in vitro but did not include this in the paper (referred to as "data not shown"). What we do not know is whether this is parylated in

vivo and whether this is a major player or whether other things might be involved. One experiment we could do to address this would be to deplete H2A.Z and then check parylation at the FOS promoter. If H2A.Z is a major target for PARP-1, then we should see loss of parylation signal. This would then imply that other things are likely not the direct targets for parylation, or at least that H2A.Z is one of the major targets.

So, as before, I feel that this paper addresses an interesting and timely subject area that builds upon growing evidence in the literature for interactions between PARP-1 and H2A.Z. But a major shortcoming remains, namely the lack of data that could confirm and elucidate in more detail the proposed mechanisms.

Response: We appreciate the referee's viewpoint here. However we suggest performing two relatively quick experiments which would help extend the study to address the points raised. These are described above but to reiterate we propose to:

[1] Use a PARP inhibitor to provide direct evidence that it is the enzymatic activity of PARP-1 that is important for controlling p400/INO80 binding kinetics.

[2] Deplete H2A.Z and then monitor whether parylation is lost at the FOS promoter.

Would these experiments be sufficient to make the paper more conclusive in terms of mechanism? We believe that our data fully supports the title, which is a major advance in our understanding of PARP-1 in transcriptional regulation.

3rd Editorial Decision

04 July 2013

Thank you for your comments in response to my decision, which are well taken. We certainly appreciate that during revision you extended your analysis to include PARP-1 dependent dynamics of p400 and INO80 occupancy at the fos promoter. However, as clearly stated in the balanced review of referee 2, your study nevertheless fails to provide definitive molecular insight into how PARP-1 functions in this context. Our initial invitation to revise your manuscript was a borderline decision, and I stressed at the time that we would require considerably more mechanistic understanding, which entailed challenging experimentation.

Given our specified expectations, and the fact that two well-known experts in the field, who are familiar with the journal and its scope, are unable to provide their essential support for publication of the manuscript in The EMBO Journal, I currently do not see sufficient reason to involve a fourth reviewer.

I am very sorry to not be able to come to a more positive conclusion and hope that you will soon receive more encouraging news elsewhere.

Additional correspondence (author)

16 July 2013

Thanks for the response and considering this. I agree that maybe it is best to stick with the reviewers we have got and deal with the issues they raise. As an Editor myself, that would be the stance I would take. As you indicated, referee 2 has a balanced viewpoint, and we have considered this carefully. They seem not to require anything extensive (ie they recognise that reconstitution experiments are too much of an ask) and we believe that we can deal with their points by two fairly simple experiments which do not in our view constitute a major revision. Our response to this reviewer and a suggestion of two experiments are added to this response. I would be grateful if you could consider this request and forward these comments to the reviewer for his/her views. Even if they are negative, it would help us gauge the degree of experimentation that might be required for submitting elsewhere.

Additional correspondence (editor)

19 July 2013

Thanks for your response, and my apology for the slight delay in getting back to you. Please let me reiterate that due to the reasons outlined in my decision letter, we will not be able to offer publication of your study at The EMBO Journal. However, given the interest of your observations and the fact that the reviewers' main concern was a shortcoming in providing definitive molecular insight, I took this opportunity to discuss your manuscript with my colleague Esther Schnapp at EMBO Reports (cc-ed here). EMBO Reports specializes on shorter papers with a high level of conceptual advance, rather than emphasizing the depth of mechanistic detail. In this regard your data set would be an excellent fit for their scope. Esther is interested in the publication of your study, although this would require you to shorten the manuscript to conform to EMBO Reports standards (5 main figures; 30,000 characters max). In addition, the suggested experiment using the PARP inhibitor to provide direct evidence that it is the enzymatic activity of PARP-1 that is important for controlling p400/INO80-binding kinetics would be very valuable. If you find this option worthwhile, it would be easiest if you discuss the details with Esther – she would be happy to help with the necessary cuts.

For The EMBO Journal, I am sorry that I do not have more positive news on this occasion, but I do hope that you will consider EMBO Reports!

Additional correspondence (author)

22 July 2013

Thanks for considering this again, and suggesting this alternative route. I have discussed this with the co-authors and we have decided that your suggestion is a good one and we will go with the EMBO Reports option. We think that it is important to get the takehome message out there and this might then stimulate other groups who work more directly on the chromatin remodellers to do the followup mechanistic studies. I'll send a more detailed and specific email to Esther.

Additional correspondence (author - to EMBO reports)

22 July 2013

This is a followup email to the one I responded to from your colleague concerning the paper Manuscript EMBOJ-2012-83512R1-Q, that we sent to EMBO J. Can I clarify that you are able to accept this in EMBO Reports without the need for further review or would we have to go through a further review process?

I also noted that Anke suggested that we include an additional experiment. Although we did suggest this to help provide more mechanistic insight for potential inclusion in EMBO J., this experiment is not essential to support the takehome message in the paper which is encapsulated in the current title. We would rather not do this experiment if at all possible (ie investigate whether PARP activity affects p400 recruitment) as the postdoc who is first author is still on maternity leave and the other postdoc has his own project underway and would have to fit this in (obviously with no guarantee of success either). As I will go on holiday in 2 weeks time, and this experiment would likely not be completed by then, this would then incur a delay in getting a final version to you until likely early September. I could otherwise prioritise this and get it back to you during the next 2 weeks and thereby speed up the process considerably.

Do you have any suggestions for manuscript shortening? It is fairly easy to move some parts of the figures to supplementary (all of Fig. 4 could potentially go for example), they would still need describing in the text which is what I find hardest to do. I guess the discussion would need compressing as well, and any suggestions would be welcome, as I do not like not putting things into context of other people's work. A lot of the materials and methods could be moved (and view as to how much?).

I look forward to hearing from you.

Additional correspondence (editor - EMBO reports)

23 July 2013

Thank you for your email and for taking up the offer to publish your manuscript in EMBO reports. I have looked at your manuscript again and my suggestions are:

Figures 1 and 2 could be combined, if only the most relevant data are shown. Confirmations in different cell lines can be moved to the supplement, for example.

Figure 4 can be moved to the supplement.

Figures 5-7 need to be kept as the most important data are here. The model could be moved to the SI. Can you please indicate in the model where p400 and Ino80 are?

Figure 3 could potentially be moved to the SI (if you do not want to combine figures 1 and 2)

I think it would be nice if the manuscript focused on the role of PARP1 in H2A.Z/H2A exchange. Our length limits are 30.000 characters (including spaces, references, figure legends) and 5 main figures. I think the the discussion section can and needs to be substantially shortened; we also offer the possibility to combine the results and discussion section which may help to eliminate some redundancy that is inevitable when discussing the same experiments twice. The materials and methods section can also be shortened and most of it can be moved to the SI. Please note, however, that the materials and methods essential for the understanding of the experiments described in the main manuscript file must be kept in the main materials and methods section.

I have not seen the supplementary figures. We usually require the supplementary figures to be directly linked to their corresponding main figure. As far as I understand you have 10 SF, and this number needs to be reduced. We usually do not allow more than 5 SF, but we can make exceptions. If the single SF are very small, some could potentially be combined.

I noticed that error bars are shown when $n=2$, which should not be the case. If $n<3$ then no error bars or statistics should be calculated. Either the error bars need to be removed or, preferably, please repeat the experiment at least 3 times and include the statistics.

Regarding the PARP1 inhibitor experiment, this is not strictly required for publication of the manuscript by EMBO reports, although it clearly would add to the study and therefore be very welcome.

Please let me know if you have any further comments or questions.

Additional correspondence (editor)

23 July 2013

Sorry, I forgot to say that the paper would not need to be reviewed again.

You can send us the files also via email if this is easier for you and we will upload them for you into our system.

Regarding the statistics, n should stand for the number of independent experiments and not for technical repeats. It does not make much sense to calculate statistics and error bars for technical repeats, since this only shows how careful the repeats were done, but not how solid the actual results/data are. I have asked Bernd Pulverer (Head of the EMBO publications) whether we can keep the error bars if $n=2$, but we usually do not allow it. I will let you know what he says.

Submission to EMBO reports

2nd August 2013

I have enclosed a manuscript for publication in *EMBO Reports* entitled "*PARP1 orchestrates variant histone exchange in signal-mediated transcriptional activation*". This is an amended version of one which was submitted to EMBO J.

I have significantly reduced the character count to nearly half of the original manuscript size and it now stands at 30,699 (including spaces, references and figure legends). I have also condensed the paper down to five figures. The supplementary is also reduced to 5 figures which generally map on to each of the figures in the manuscript. We have also carefully gone through all the figure legends to make sure it is clear what the bars on the figures and error bars indicate. Where "n" is stated this always now refers to biologically independent experiments. We have included all the data from the original paper with the exception of previous Fig. 1A which was internally redundant with another subpart of the same figure that is retained. Thus, the paper represents the same data as was previously reviewed for EMBO J. We decided not to add any further experiments for reasons of time and also space in what is already a very substantial paper. As indicated previously, the additional suggested experiment is not in our view essential for the overall takehome message of the paper. Please let me know if you need anything else before publication can be finalized in EMBO Reports. I look forward to receiving your final decision.

Additional correspondence (editor)

7 August 2013

Thank you for sending the shortened version of your manuscript. I think the text reads well, and in principle your manuscript could be accepted now. However, the statistical analyses and data representation need to be improved.

I saw now that I was not very clear on this in my last email, and I apologize. We usually do not allow authors to show error bars if $n=2$, as statistical analyses can be misleading for very small sample size. We strongly encourage authors to perform at least three independent experiments and show the averages of these with error bars. In most of your figure legends it reads "average of at least 2 independent experiments" and it is not clear what this means. Where is $n=3$ and where is $n=2$? Can you please explain this for each figure panel, including the supplementary figures? Also, in the supplementary figures, it seems that n does not represent the number of experiments, as you write, for example, $n=2$ but then 4 independent experiments were performed. Can you please go carefully through the text and specify for each legend how many experiments were performed and use n as the number for independently performed experiments, and not for replicates of the same experiment.

If you want to show error bars when $n=2$ then the single data points of the two independent experiments must be shown, either in the source data (that will be linked to the figure online) or in the main figure panel, along with the averages and error bars. This is the first time that I allow authors to show error bars although $n=2$, and I have to say that I am not comfortable this way, also, because it seems that for most of your figure panels, $n=2$ only, which is really suboptimal.

Some of the (supplementary) figure panels currently do not specify what the bars and error bars represent, please include this missing information in the legends.

For more information and good practice on statistics and error bars in molecular biology, please see the two articles that I attached to this email.

I am looking forward to receiving the revised manuscript and supplementary information (and may be source data) as soon as possible, and please let me know if you have any further questions.

Additional correspondence (author)

19 August 2013

Sorry for the delay in response. I have now attached a revised main manuscript and supplementary PDF. I also have provided a detailed explanation about our use of error bars when calculating averages. I think we have had crossed wires somewhere, as we generally do more than 3 independent biological replicates, involving experiments done on at least two independent occasions. In the previous version I was defining "n" as numbers of different occasions done rather than number of biological replicates, which is now the definition applied to "n" in the figure legends. You will note that we have not discussed statistical values in the manuscript (ie P-values) as we agree that this is not valid where $n=2$ (but we could have done this for the majority of the figures where $n \geq 3$).

We still have a few experiments in there with $n=2$. Personally, I think this is still okay, as we make it pretty clear what the error bar consists of and shows the spread of the two points. Importantly we do not apply any statistical tests to this data. In the majority of cases, where $n=2$, the experiments are done on separate days rather than being parallel independent biological samples processed on the same day. The question now is whether you want us to do any more about the issue where we only have two replicates. I think that showing all the data points would be very cumbersome (especially as this is effectively the point of the error bars). To link to the source data, what would you suggest we would do (if you still think this is necessary) ie what would you expect? Presumably a simplified Excel spreadsheet?

Personally, as a journal Editor myself, who deals with these things all the time, I am happy with people who take the same approach as we have done (which is the majority of people- in fact most people do not even include experiments done on different days but just biological replicates on the same day). However, I do appreciate the points you are raising, and it is very important to make things clear as to what the data represent (which I think we have now done).

We await further advice.

Additional correspondence (editor)

21 August 2013

Thank you for the explanations, I think I understand now what the data represent.

However, I have to admit that I find the figure legends rather confusing, stating "n" and then a different number of independent experiments and the technical replicates. I discussed this with my colleagues and also with our manager of the EMBO publications, Bernd Pulverer, and we agree that it would be better to combine the number of biologically independent experiments done on the same day and done on different days, and define this as "n" and add this information to the main methods section. You could also explain once in the main method section that single data points for ChIP and qPCR experiments were calculated from 2 or more technical replicates, and this information could then be deleted from the figure legends. The specifications of "n" and the error bars need to be kept in the legends though.

For the few cases where $n=2$, we remain of the opinion that it is very important to show the single data points, either in the graphs (along with the means and error bars if you want), or as source data, for example as a simplified Excel spreadsheet, as you suggest. The source data will then be linked to the main figure online.

Regarding the presentation of "representative data" of a larger pool of total experiments, please add the information that the means and error bars are calculated from representative data from x out of $n=y$ experiments to the figure legend. Or just use $n=x$ (and add source data if $x=2$).

I hope you agree with our suggestions, and I am sorry that we did not manage to clarify these points right away.

I am looking forward to receiving the final version of the figure and legends.

Additional correspondence (author)

6 September 2013

I have attached the changed and new files you need (all the rest stay the same). I added a paragraph to the Materials and methods about the data analysis with the definition of "n". All the figure legends use n in this sense, so are already all correct. Where $n=2$, we have now added the source data in a single Excel file. Please let me know if there is anything else that you need.

Editorial Decision (EMBO reports)

16 September 2013

I am very pleased to accept your manuscript for publication in the next available issue of EMBO reports. Thank you for your contribution to our journal.